# NuMA recruits dynein activity to microtubule minus-ends at mitosis

Christina L Hueschen[1,2], Samuel J Kenny[3], Ke Xu[3], Sophie Dumont[1,2,4*]

[1]Department of Cell and Tissue Biology, University of California, San Francisco, San Francisco, United States; [2]Biomedical Sciences Graduate Program, University of California, San Francisco, San Francisco, United States; [3]Department of Chemistry, University of California, Berkeley, Berkeley, United States; [4]Department of Cellular and Molecular Pharmacology, University of California, San Francisco, San Francisco, United States

**Abstract** To build the spindle at mitosis, motors exert spatially regulated forces on microtubules. We know that dynein pulls on mammalian spindle microtubule minus-ends, and this localized activity at ends is predicted to allow dynein to cluster microtubules into poles. How dynein becomes enriched at minus-ends is not known. Here, we use quantitative imaging and laser ablation to show that NuMA targets dynactin to minus-ends, localizing dynein activity there. NuMA is recruited to new minus-ends independently of dynein and more quickly than dynactin; both NuMA and dynactin display specific, steady-state binding at minus-ends. NuMA localization to minus-ends involves a C-terminal region outside NuMA's canonical microtubule-binding domain and is independent of minus-end binders γ-TuRC, CAMSAP1, and KANSL1/3. Both NuMA's minus-end-binding and dynein-dynactin-binding modules are required to rescue focused, bipolar spindle organization. Thus, NuMA may serve as a mitosis-specific minus-end cargo adaptor, targeting dynein activity to minus-ends to cluster spindle microtubules into poles.
DOI: https://doi.org/10.7554/eLife.29328.001

*For correspondence:
sophie.dumont@ucsf.edu

**Competing interests:** The authors declare that no competing interests exist.

## Introduction

Each time a cell divides, molecular motors help remodel the microtubule cytoskeleton into a bipolar assembly of microtubules called the spindle. The spindle's bipolar architecture is essential to its function – accurate chromosome segregation. In mammalian spindles, microtubule plus-ends mechanically couple to chromosomes, while microtubule minus-ends are focused into two poles that dictate where chromosomes are transported at anaphase. To build the spindle's architecture, motors must exert spatially regulated forces on microtubules. Microtubule ends offer platforms for such localized regulation, since they are structurally distinct. Indeed, motor recruitment and activity at plus-ends is well-documented (*Wu et al., 2006*), but motor regulation at minus-ends is less well understood.

Dynein is a minus-end-directed motor which slides parallel spindle microtubules to focus their minus-ends into spindle poles (*Heald et al., 1996*; *Verde et al., 1991*), working in complex with its adaptor dynactin and the microtubule-binding protein NuMA (*Gaglio et al., 1996*; *Merdes et al., 1996*). The clustering of parallel microtubules into poles presents a geometric problem when forces are indiscriminately applied all along microtubules: inversely oriented motors between parallel microtubules will oppose each other, resulting in gridlock, unless symmetry is broken by dynein enrichment at microtubule minus-ends (*Hyman and Karsenti, 1996*; *McIntosh et al., 1969*; *Surrey et al., 2001*). In computational models, localizing a minus-end-directed motor at microtubule ends permits microtubule clustering into asters or poles (*Foster et al., 2015*; *Goshima et al., 2005*; *Nedelec and Surrey, 2001*; *Surrey et al., 2001*) and the emergence of a robust steady-state spindle length (*Burbank et al., 2007*). More recently, experimental work has shown that dynein-dynactin

**eLife digest** Every time a cell divides, it needs to duplicate its DNA and evenly distribute it between the two new 'daughter' cells. To move and distribute DNA, the cell builds a large machine called a spindle, which is made of stiff cables called microtubules.

Many proteins, including a motor called dynein, help to organize the spindle's microtubules. One of dynein's jobs is to cluster all microtubules at the two tips of the spindle, pulling them into shape. Without this clustering, spindles have the wrong shape and structure and can make mistakes when moving DNA.

Microtubules have a 'plus' end and a 'minus' end, and motor proteins usually only travel in one specified direction. Dynein, for example, moves towards the minus end of microtubules, which is where most of the clustering happens. It can form a complex with other proteins that help clustering, one of which is called NuMA. Until now, it was thought that dynein transports NuMA to the minus ends.

To test this model, Hueschen et al. studied dividing human cells under a microscope and isolated minus ends with the help of a laser. The experiments showed that, in fact, NuMA gets to minus ends independently of dynein. Once it is on the minus ends, NuMA grabs hold of another protein complex called dynactin, which then gathers dynein. Dynein then pulls the spindles into shape from the minus ends. When NuMA was experimentally removed from the cells, dynein-dynactin complexes were scattered along the entire length of the microtubule instead of pulling specifically on minus-ends, which resulted in disorganized spindles. Thus, where dynein complexes pull determines what spindle shape they build.

Hueschen et al. also showed that dynein complexes only pull on minus-ends while the cell divides, which makes sense, because NuMA remains hidden in the cell nucleus for the rest of the time. Together, these results suggest that NuMA makes sure dynein pulls specifically on the minus-ends of the microtubules to tighten the spindle at the right time.

A next step will be to test how the mechanical properties of the spindle are changed without dynein pulling on minus-ends. A better knowledge of how different proteins work together to build the spindle and help cells divide can help us understand what goes wrong when cells divide abnormally, as in the case of cancer cells.

DOI: https://doi.org/10.7554/eLife.29328.002

and NuMA do indeed selectively localize to spindle minus-ends, with dynein pulling on them after kinetochore-fiber (k-fiber) ablation in mammalian spindles (*Elting et al., 2014*; *Sikirzhytski et al., 2014*). This is consistent with suggestions that dynein and NuMA capture and pull on distal k-fiber minus-ends in monopolar spindles (*Khodjakov et al., 2003*). Altogether, these findings demonstrate the importance (in silico) and existence (in vivo) of localized dynein activity at spindle microtubule minus-ends.

How dynein becomes localized at minus-ends remains an open question. Dynein may be enriched near minus-ends because it walks toward them on microtubules and piles up when it runs out of track; indeed, pile-up of dynein has been observed at minus-ends in vitro (*McKenney et al., 2014*; *Soundararajan and Bullock, 2014*) and can drive minus-end clustering (*Tan et al., 2017*). Alternatively, the exposed α-tubulin interface at minus-ends is structurally distinct and could bind an adaptor protein that specifically recruits dynein, analogous to recruitment at canonical dynein cargoes like organelles (*Kardon and Vale, 2009*). NuMA can target dynein-dynactin to the cell cortex (*Lechler and Fuchs, 2005*; *Nguyen-Ngoc et al., 2007*) and thus could be one such adaptor. However, in vitro NuMA has shown no direct affinity for minus-ends specifically, binding all along the microtubule lattice (*Du et al., 2002*; *Forth et al., 2014*; *Haren and Merdes, 2002*) or at both ends (*Seldin et al., 2016*), unlike three proteins known to interact directly with minus-ends at mitosis: γ-TuRC (*Zheng et al., 1995*), CAMSAP1 (*Akhmanova and Hoogenraad, 2015*; *Hendershott and Vale, 2014*; *Jiang et al., 2014*) and KANSL1/3 (*Meunier et al., 2015*). In cells, NuMA is thought to require dynein activity to carry it to minus-ends and spindle poles (*Merdes et al., 2000*), where it anchors spindle microtubules (*Dionne et al., 1999*; *Gaglio et al., 1995*; *Heald et al., 1997*; *Silk et al., 2009*). Thus, it remains unclear whether dynein-dynactin and NuMA have specific binding

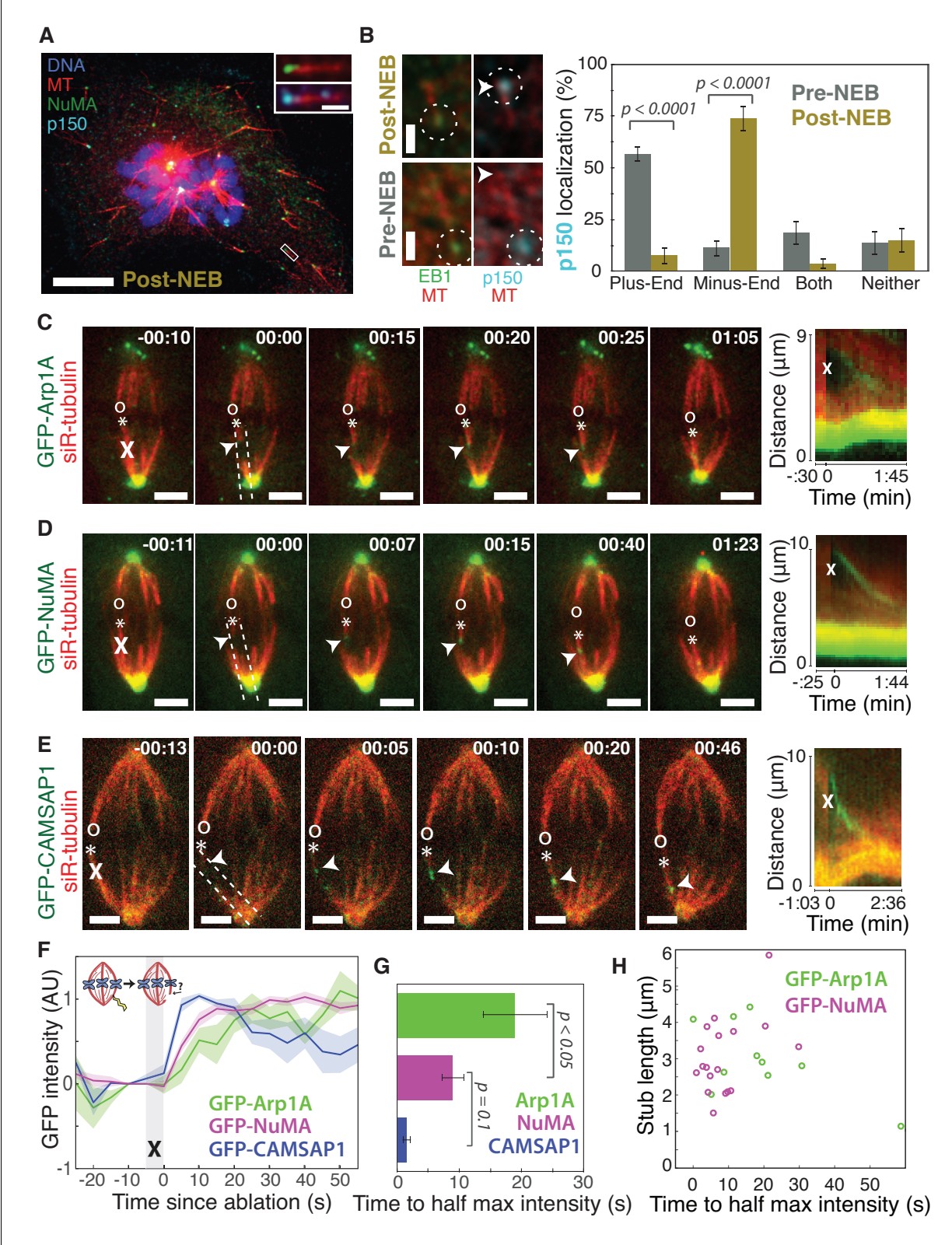

**Figure 1.** Dynactin and NuMA display specific, steady-state binding at mitotic minus-ends. See also *Figure 1—figure supplement 1* and *Videos 1– 3*. (A) Representative immunofluorescence image showing co-localization of NuMA (green) and p150 (dynactin subunit; cyan) at microtubule minus-ends in mitotic PtK2 cells (post-NEB) fixed after washout of 5 µM nocodazole. Scale bar, 10 µm. Inset: zoom of white box, with 1 µm scale bar. (B) Representative immunofluorescence images of mitotic RPE1 cells, processed as in (A). After nuclear envelope breakdown (post-NEB), EB1 (green) and

*Figure 1 continued on next page*

*Figure 1 continued*

p150 (cyan) localize to opposite microtubule ends. In prophase cells (pre-NEB), p150 instead co-localizes with EB1 at plus-ends. Dashed white circles highlight ends. Scale bar, 1 μm. Graph displays mean percentage ±SEM of p150 at each location within one cell for *n* = 72 microtubules, 8 cells (pre-NEB); *n* = 57 microtubules, 16 cells (post-NEB). (C–E) Representative time-lapse live images and kymographs of PtK2 cells expressing (C) GFP-Arp1A, (D) GFP-NuMA, or (E) GFP-CAMSAP1. Arp1A, NuMA, and CAMSAP1 are all recruited to k-fiber minus-ends (arrowheads) created by ablation ('X's) and move with them as dynein pulls them poleward (*Elting et al., 2014*). Time is in min:s, with the frame captured immediately following ablation set to 00:00. '*' marks plus-end of ablated k-fiber, and 'o' marks its sister. Scale bar, 5 μm. Kymographs (right) are along poleward paths (dashed lines). (F) Plot of mean normalized GFP intensity and SEM (shading) over time at ablation-created minus-ends for GFP-Arp1A, GFP-NuMA, and GFP-CAMSAP1. Time = 0 s at the first frame following ablation. *n* = 10 ablations, 7 cells (Arp1A); *n* = 18 ablations, 7 cells (NuMA); *n* = 13 ablations, 7 cells (CAMSAP1). (G) Time from ablation to half maximum GFP intensity, calculated for each individual ablation (see Materials and methods) and then averaged for data in (F). Error bars show SEM. p-Values calculated by Tukey post hoc test after one-way ANOVA ($F_{(2,38)}$ = 9.26, p=0.0005). (H) Time to half-maximum GFP-Arp1A or GFP-NuMA intensity at ablation-created minus-ends as a function of length of ablation-created k-fiber stubs. Correlation coefficient = −0.62, p=0.06, *n* = 10 (Arp1A); correlation coefficient = 0.44, p=0.07, *n* = 18 (NuMA).

DOI: https://doi.org/10.7554/eLife.29328.003

The following figure supplement is available for figure 1:

**Figure supplement 1.** Dynactin localization shifts from plus-ends to minus-ends upon nuclear envelope breakdown.

DOI: https://doi.org/10.7554/eLife.29328.004

sites at minus-ends, and if so, whether they are recruited by known minus-end binders. Finally, knowing how dynein is targeted to minus-ends would make it possible to test the in vivo role of minus-end-localized – compared to indiscriminately-localized – forces in spindle organization.

Here, we use laser ablation in cells to create new, isolated minus-ends in the mammalian spindle, and quantitative imaging to map protein recruitment to these ends and its mechanistic basis. We demonstrate that NuMA binds at minus-ends independently of dynein, and that NuMA targets dynactin – and thereby dynein activity – to spindle minus-ends. This challenges the prevailing model that dynein delivers NuMA to spindle minus-ends and poles (*Merdes et al., 2000*; *Radulescu and Cleveland, 2010*). NuMA localization to minus-ends is independent of known direct minus-end binders γ-TuRC, CAMSAP1, and KANSL1/3, and it involves both NuMA's canonical microtubule-binding domain and an additional region of its C-terminus. Thus, NuMA – which is sequestered in the nucleus at interphase (*Lydersen and Pettijohn, 1980*) – may serve as a mitosis-specific minus-end cargo adaptor, recruiting dynein activity to spindle minus-ends. Both NuMA's minus-end-binding domain and dynactin-binding domain are required for correct spindle architecture, supporting long-standing in silico predictions that localizing dynein to minus-ends enables effective clustering of parallel microtubules into poles. These findings identify a mechanism for mitosis-specific recruitment of dynein to microtubule minus-ends and, more broadly, illustrate how spatial regulation of local forces may give rise to larger scale cytoskeletal architectures.

## Results

### Dynactin and NuMA display mitosis-specific, steady-state binding at microtubule minus-ends

To visualize the spatial targeting of the dynein-dynactin complex to microtubule minus-ends – which are normally buried in dense mammalian spindles – we used nocodazole washout and laser ablation to create resolvable minus-ends in mitotic cells. First, to determine whether dynein-dynactin and NuMA localize to individual microtubule minus-ends, we treated mammalian PtK2 and RPE1 cells with the microtubule-depolymerizing drug nocodazole and fixed cells 6–8 min after drug washout to capture acentrosomal microtubules with clearly visible plus- and minus-ends (*Figure 1A*). p150$^{Glued}$ (p150, a dynactin subunit) and NuMA strongly co-localized at one end of these individual microtubules (*Figure 1A*), with a clear binding preference for minus-ends over the microtubule lattice or the plus-end when polarity was marked by EB1 (*Figure 1B*). Interestingly, in prophase cells before nuclear envelope breakdown, p150 localized predominantly to plus-ends rather than minus-ends (*Figure 1B*; *Figure 1—figure supplement 1*), consistent with dynactin's interphase localization (*Vaughan et al., 1999*). Thus, nuclear envelope breakdown (NEB) confers dynactin's preference for minus-ends, suggesting regulated, mitosis-specific spatial targeting.

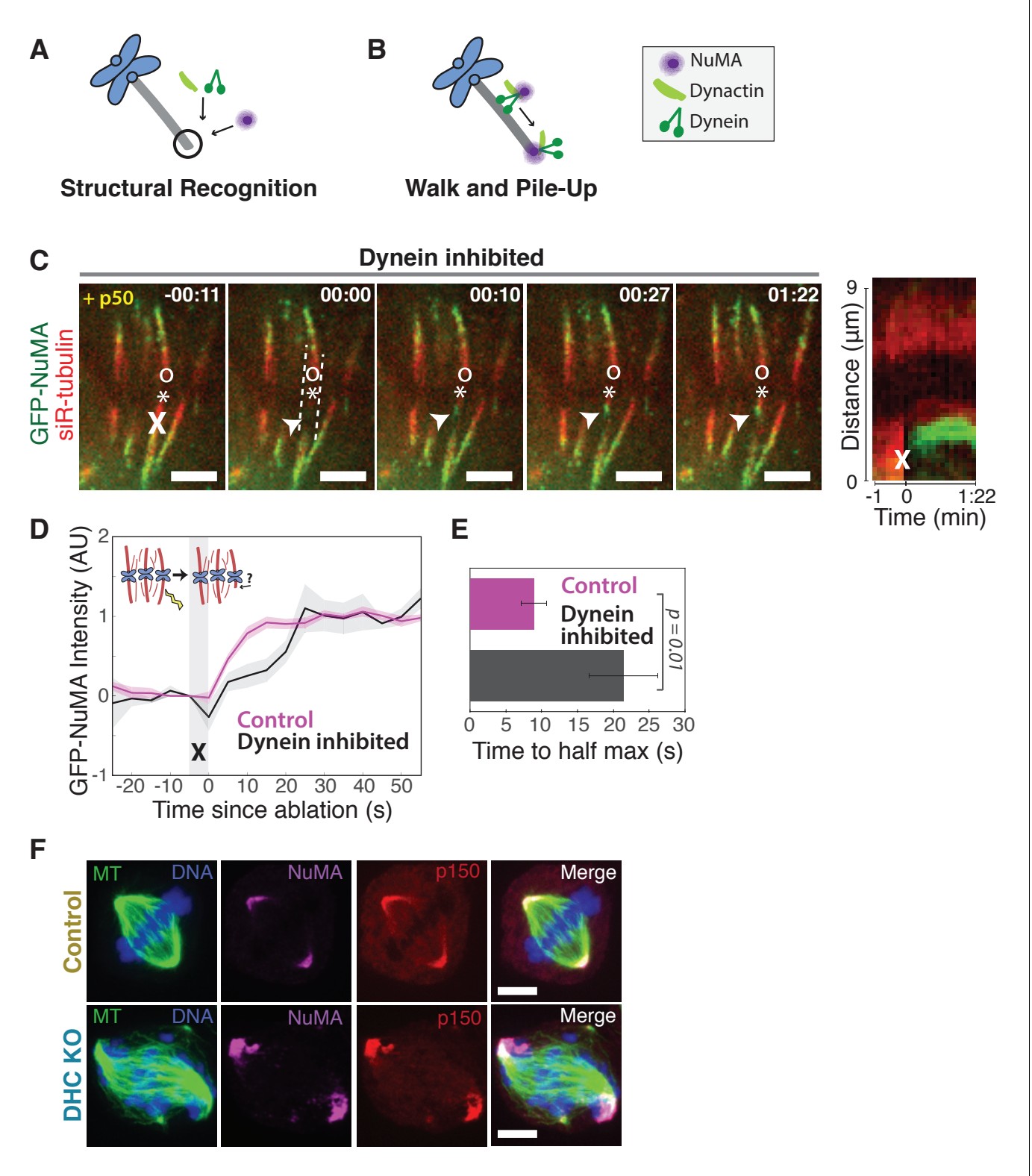

**Figure 2.** NuMA can localize to minus-ends independently of dynein. See also *Figure 2—figure supplement 1* and *Video 4*. (A–B) Models for targeting dynein to minus-ends. Model A, Structural Recognition: Cytoplasmic molecules may recognize the exposed α-tubulin interface at minus-ends and recruit dynein. Model B, Walk and Pile-up: Minus-end-directed dynein may walk to minus-ends and pile up. (C) Representative time-lapse live images and kymograph of PtK2 cell expressing GFP-NuMA, in which dynein-dynactin is inhibited by overexpression of p50 (dynamitin). K-fibers are unfocused and splayed, but NuMA is still robustly recruited to k-fiber minus-ends (arrowheads) created by ablation ('X'). Time is in min:s, with frame

*Figure 2 continued on next page*

*Figure 2 continued*
captured immediately following ablation set to 00:00. '*' marks plus-end of ablated k-fiber, and 'o' marks its sister. Scale bars, 5 μm. Kymograph (right) taken along dashed line path. (D) Plot of mean normalized GFP-NuMA intensity and SEM (shading) over time at ablation-created minus-ends. Time = 0 s at the first frame following ablation. n = 18 ablations, 7 cells for control; n = 14 ablations, 8 cells for dynein-inhibited (p50 overexpression). (E) Time from ablation to half maximum GFP-NuMA intensity, calculated for each individual ablation (see Methods) and then averaged for data in (D). Error bars show SEM. (F) Representative immunofluorescence images of inducible-Cas9 dynein heavy chain (DHC)-knockout HeLa cells (*McKinley and Cheeseman, 2017*) showing robust localization of NuMA and p150 (dynactin) at minus-ends when DHC is deleted. Scale bar, 5 μm.
DOI: https://doi.org/10.7554/eLife.29328.008
The following figure supplement is available for figure 2:

**Figure supplement 1.** NuMA localizes to minus-ends despite dynein inhibition or knockout.
DOI: https://doi.org/10.7554/eLife.29328.009

Second, we sought to test whether dynactin and NuMA have finite binding sites at minus-ends by measuring the kinetics of their recruitment. To do so, we used laser ablation of k-fibers in PtK2 cells to create new minus-ends within the spindle body (*Figure 1C–E*). By spatially and temporally synchronizing the creation of a bundle of minus-ends, laser ablation allowed for dynamic measurements of the recruitment to minus-ends of GFP-tagged dynactin (Arp1A) and NuMA and comparison to a direct minus-end binding protein, CAMSAP1. High dynein background in the cytoplasm prevented accurate recruitment kinetics measurements of the motor itself. Dynactin, NuMA, and CAMSAP1 robustly recognized new microtubule minus-ends within the spindle (*Figure 1C–E*; *Videos 1–3*). The kinetics of dynactin and NuMA recruitment to minus-ends were distinct from CAMSAP1, which reached a max intensity after approximately 5–10 s, but then decreased in intensity as if its binding sites at the minus-end were gradually obstructed (*Figure 1E–G*). Unlike CAMSAP1, dynactin and NuMA intensities reached a stable plateau, suggesting that binding saturates, reaching a steady-state. This finite binding is consistent with a truly end-specific localization (rather than localization along the unbounded lattice *near* the minus-end). In addition, the rate of NuMA or dynactin accumulation at new minus-ends did not correlate with the length of k-fiber stubs created by ablation (*Figure 1H*), which could indicate that recruitment rate is set by the number of individual microtubule minus-ends (which is similar across k-fibers [*McEwen et al., 1997*]) rather than k-fiber length.

Interestingly, NuMA intensity at new minus-ends increased at a faster rate and saturated sooner than dynactin (*Figure 1F,G*). This observation hints at a model in which NuMA targets dynein-dynactin to minus-ends, and it is less easily consistent with the idea that dynein delivers NuMA to minus-ends. If true, this hypothesis would explain why dynactin's minus-end preference arises after NEB, when NuMA is released from the nucleus into the mitotic cytoplasm.

## NuMA localizes to minus-ends independently of dynein

The finding that NuMA localizes to minus-ends more quickly than dynactin could suggest that NuMA directly or indirectly recognizes the exposed α-tubulin structure of minus-ends and subsequently recruits dynein-dynactin ('Structural Recognition,' *Figure 2A*). Alternatively, dynein-dynactin could walk toward minus-ends, carrying NuMA, and pile up or dwell there ('Walk and Pile-up,' *Figure 2B*). To test the 'Walk and Pile-up' model, we inhibited dynein-dynactin in PtK2 spindles by over-expressing p50 (dynamitin), resulting in fully unfocused k-fiber arrays (*Figure 2C*). The 'Walk and Pile-up' model predicts that in the absence of dynein-dynactin transport, NuMA – which requires dynactin for its interaction with dynein (*Merdes et al., 2000*) – should not reach minus-ends. Instead, GFP-NuMA was robustly recruited to minus-ends created by k-fiber ablation (*Figure 2C*; *Video 4*), indicating that NuMA can localize to minus-ends independently of dynein activity. Similarly, NuMA was recruited to ablation-created minus-ends when dynein was inhibited by p150-CC1 over-expression (*Figure 2—figure supplement 1A,B*). Thus, NuMA localizes to minus-ends without dynein carrying it there – consistent with early observations in extract asters and spindles (*Gaglio et al., 1996*; *Heald et al., 1997*) but in contrast to the prevailing view that dynein delivers NuMA to minus-ends (*Merdes et al., 2000*; *Radulescu and Cleveland, 2010*). Slower NuMA accumulation kinetics after p50 overexpression suggest that dynein-dynactin-NuMA complex formation may aid rapid NuMA recruitment, but dynein 'Walk and Pile-up' alone cannot explain NuMA's minus-end affinity (*Figure 2D,E*).

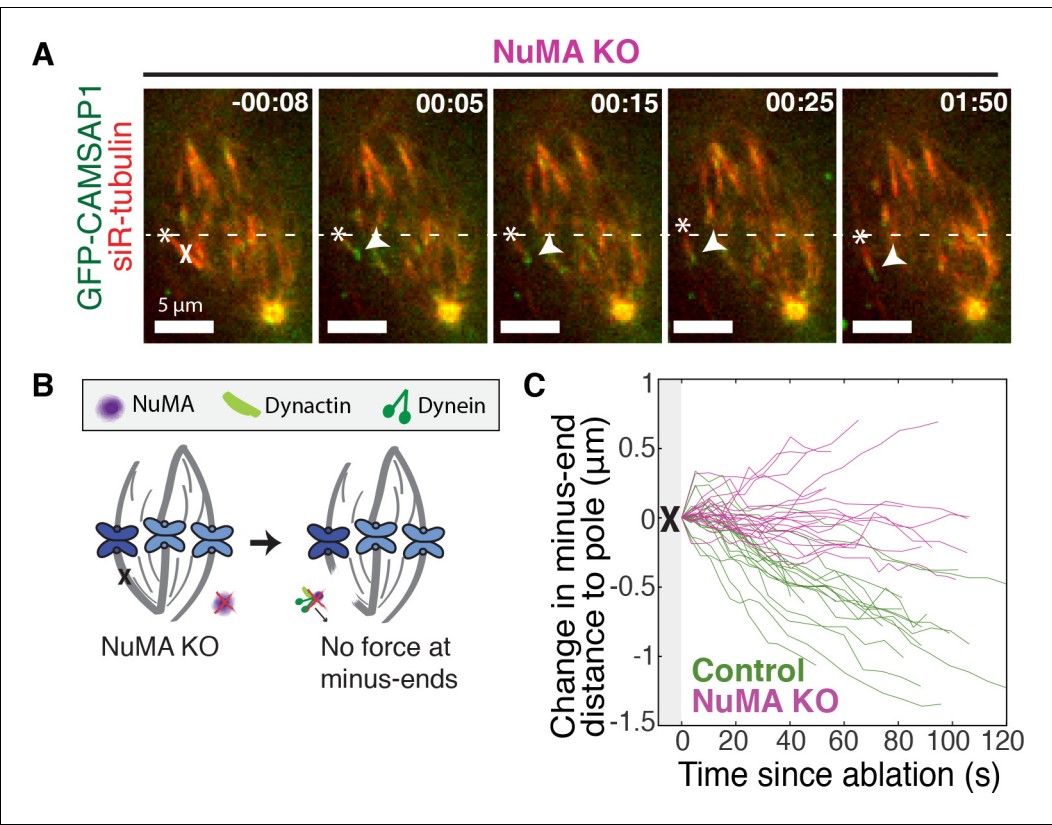

**Figure 3.** NuMA is required for force generation at minus-ends. See also *Figure 3—figure supplement 1* and *Video 5*. (A) Representative time-lapse live images of RPE1 cell in which NuMA has been knocked out using an inducible Cas9 system. GFP-CAMSAP1 is expressed to label minus-ends. After ablation ('X'), the k-fiber minus-end (arrowhead) is not transported poleward by dynein and remains detached from the spindle. Time is in min:s, with the frame captured immediately following ablation set to 00:00 s. Plus-end of ablated k-fiber is marked by '*'. Scale bars, 5 µm. (B) Cartoon: In the absence of NuMA, force generation at ablation-created minus-ends is not observed. (C) Movement of ablation-created minus-ends (marked by GFP-CAMSAP1) relative to the pole. In control cells (green traces), minus-ends are transported toward the pole by dynein at consistent speeds, but this transport is lost when NuMA is knocked out (purple traces). n = 19 ablations, 8 cells (control); n = 20 ablations, 6 cells (NuMA knockout).

DOI: https://doi.org/10.7554/eLife.29328.011

The following figure supplement is available for figure 3:

**Figure supplement 1.** Inducible CRISPR/Cas9 NuMA knockout.

DOI: https://doi.org/10.7554/eLife.29328.012

To confirm that NuMA localizes to minus-ends even after genetic dynein deletion, we used an inducible CRISPR/Cas9 HeLa cell line to knock out dynein's heavy chain (DHC) (*McKinley and Cheeseman, 2017*). After dynein knockout (*Figure 2—figure supplement 1C,D*), both NuMA and dynactin localized robustly to k-fiber minus-ends (*Figure 2F*; *Figure 2—figure supplement 1E,F*). Together, these data indicate that NuMA can bind to minus-ends, either directly or indirectly, in the absence of dynein and are consistent with NuMA-mediated recognition of minus-ends ('Structural Recognition,' *Figure 2A*). In addition, they suggest that NuMA may interact with and target dynactin to minus-ends even in the absence of dynein, when dynein-dynactin complexes cannot form and dynein motility cannot deliver dynactin close to minus-ends.

## NuMA is required for dynein activity and dynactin localization at minus-ends

If NuMA does target dynein-dynactin to minus-ends, localizing force there, we would expect NuMA to be required for dynein to transport minus-ends created by k-fiber ablation (*Elting et al., 2014*;

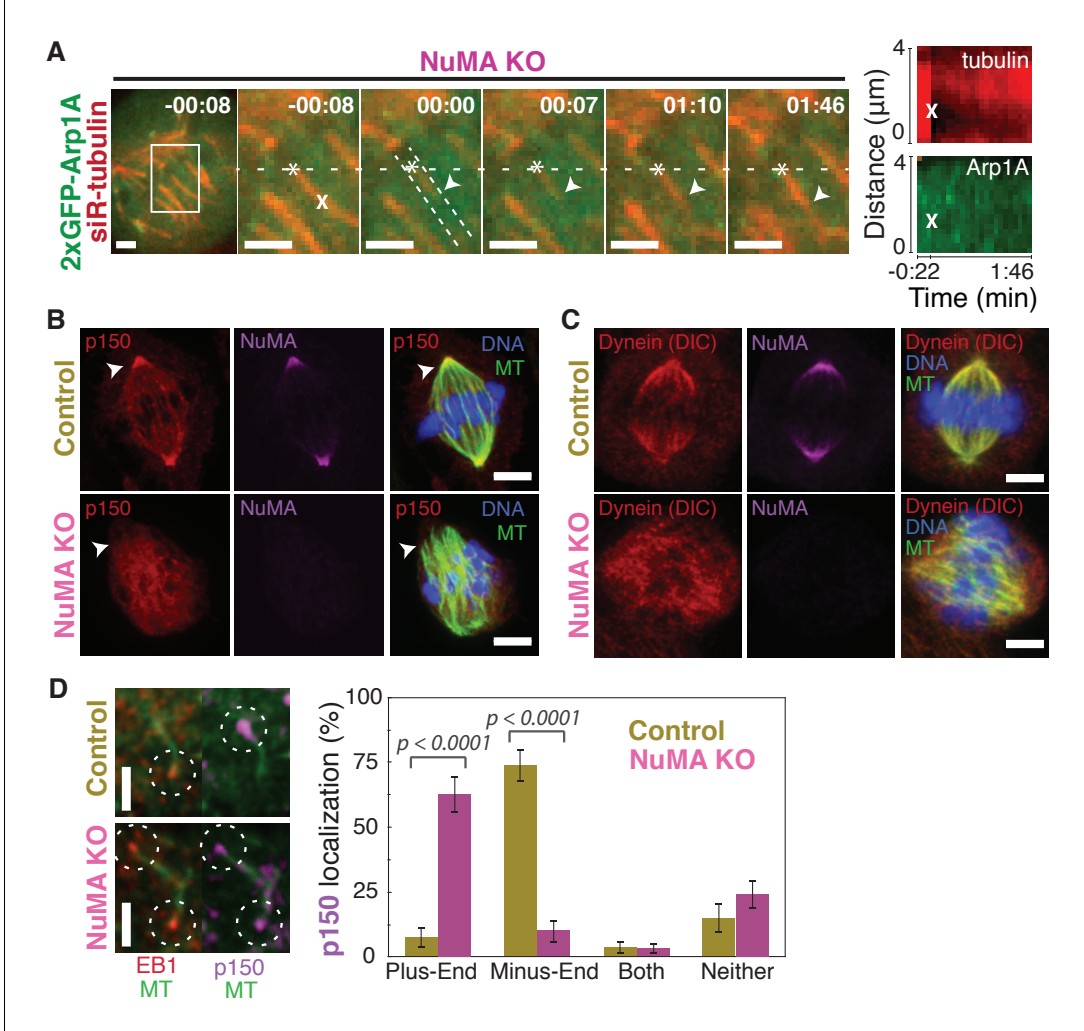

**Figure 4.** NuMA recruits dynactin to mitotic minus-ends. See also *Video 6*. (A) Representative time-lapse live images and kymograph of RPE1 cell in which NuMA has been knocked out using an inducible Cas9 system. 2xGFP-Arp1A recruitment was never observable at minus-ends (arrowheads) created by ablation ('X') (*n* = 16 ablations, 6 cells). Time is in min:s, with frame captured immediately following ablation set to 00:00. Plus-end of ablated k-fiber is marked by '*'. Scale bars, 2 μm. Kymographs (right) taken along dashed line path. (B) Representative immunofluorescence images of control and NuMA-knockout RPE1 spindles, showing a loss of p150 (dynactin) at minus-ends in the absence of NuMA. Scale bars, 5 μm. (C) Representative immunofluorescence images of control and NuMA-knockout RPE1 spindles. Dynein localizes along spindle microtubules in both conditions; its distribution is less noticeably altered by NuMA loss. Scale bars, 5 μm. (D) Representative immunofluorescence images of mitotic RPE1 cells (post-NEB) fixed after washout of 5 μM nocodazole, as in *Figure 1A,B*. In control cells, p150 (dynactin; purple) localizes to microtubule minus-ends, opposite EB1 (red). When NuMA is knocked out, p150 co-localizes with EB1 at plus-ends. Dashed white circles highlight ends. Scale bars, 2 μm. Graph displays mean percentage ±SEM of p150 at each location within each cell, for *n* = 72 microtubules, 8 cells (control); *n* = 72 microtubules, 11 cells (NuMA knockout).
DOI: https://doi.org/10.7554/eLife.29328.014

*Sikirzhytski et al., 2014*). To test this hypothesis, we made inducible CRISPR/Cas9 RPE1 cell lines to knock out NuMA (*Figure 3—figure supplement 1*) (*McKinley et al., 2015*; *Wang et al., 2015*). Indeed, when NuMA was knocked out – causing elongated, heterogeneously disorganized spindles with detached centrosomes – ablation-created minus-ends were no longer transported toward poles by dynein (*Figure 3A–C*; *Video 5*). These data are consistent with a previous finding that NuMA antibody injection prevents distal k-fiber looping events in monopolar spindles, in which dynein likely pulls on free k-fiber minus-ends (*Khodjakov et al., 2003*). Thus, NuMA is required for dynein activity at spindle microtubule minus-ends.

Dynein activity at minus-ends could require NuMA because NuMA modulates dynein-dynactin's ability to pull on microtubules, or because NuMA localizes dynein to minus-ends. Given the findings in

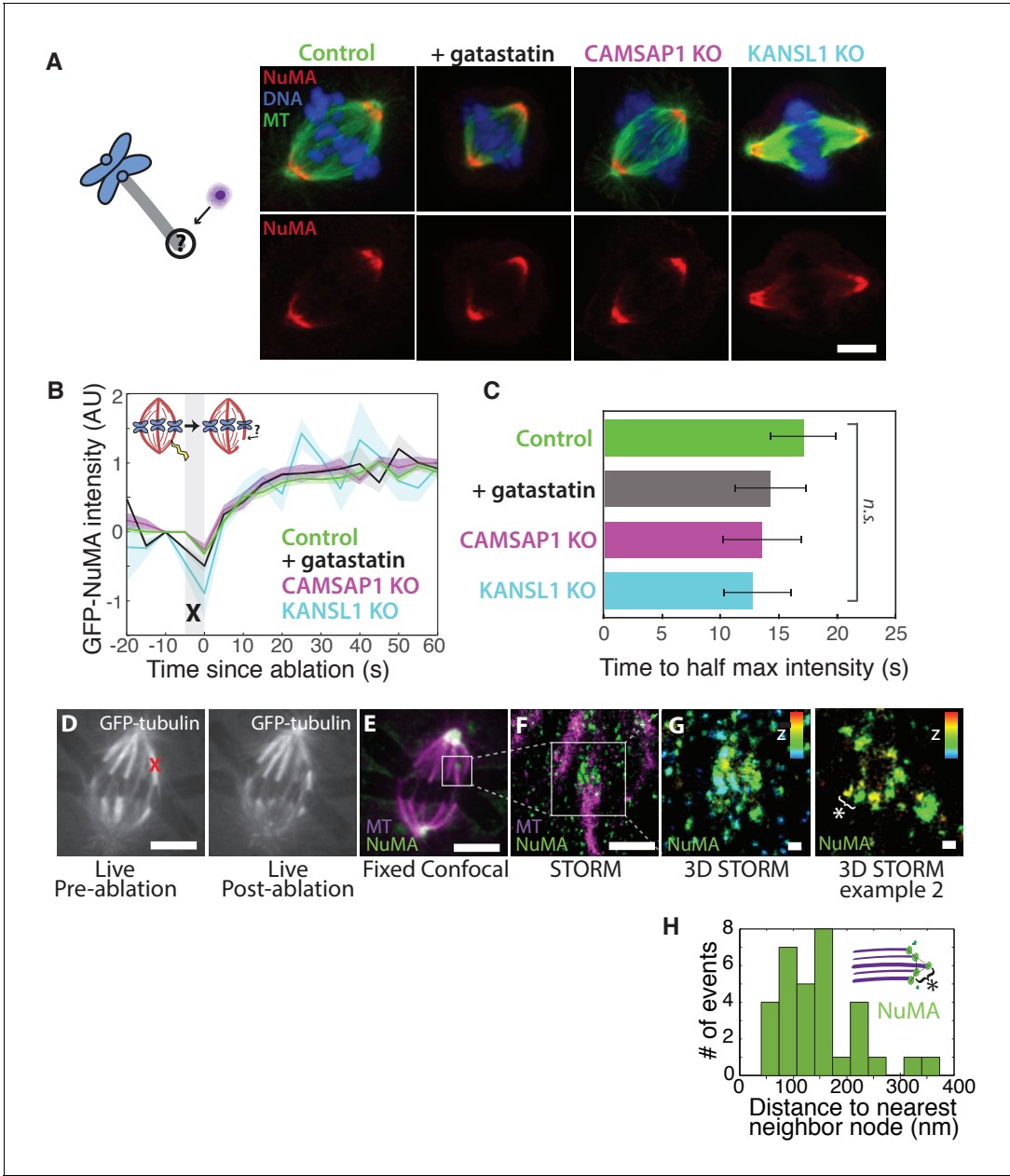

**Figure 5.** NuMA localizes to minus-ends independently of known minus-end binding proteins. See also *Figure 5—figure supplement 1*. (**A**) Schematic of hypothesis that a minus-end binding protein recruits NuMA. Instead, representative immunofluorescence images show unchanged NuMA localization in control RPE1 cells and RPE1 cells in which direct mitotic minus-end binders are inhibited (30 µM gatastatin to inhibit γ-tubulin) or knocked out (CAMSAP1, KANSL1). Scale bar, 5 µm. (**B**) Plot of mean normalized GFP-NuMA intensity and SEM (shading) over time at ablation-created minus-ends. Time = 0 s at the first frame following ablation. n = 18 ablations, 7 cells (control); n = 9 ablations, 8 cells (+gatastatin); n = 10 ablations, 5 cells (CAMSAP1 knockout); n = 8 ablations, 4 cells (KANSL1 knockout). (**C**) Time from ablation to half maximum GFP-NuMA intensity was calculated for each individual ablation (see Materials and methods) and then averaged for data in (**B**). Error bars show SEM. 'n.s.' indicates no statistically significant difference. One-way ANOVA: $F(3,42) = 0.45$, p=0.72. (**D–G**) 3D STORM of NuMA at PtK2 k-fiber minus-ends created by ablation. (**D**) K-fibers were cut using ablation (red 'X'), and cells were immediately fixed for immunofluorescence. Individual ablated spindles were imaged by spinning-disk confocal (**E**) and then by 3D STORM (**F,G**). In two right-most panels (**G**), structures are colored according to position in the Z-axis (red = +300 nm, blue = −300 nm). Scale bars: 5 µm in (**D,E**); 1 µm in (**F**); 100 nm in (**G**). (**E**) Distances between neighboring 'nodes' of NuMA (e.g., '*' here and in (**G**)) are ~50–150 nm, consistent with measured spacing between individual microtubules within PtK2 k-fibers (*McDonald et al., 1992*). n = 32 nodes, 4 ablations.
DOI: https://doi.org/10.7554/eLife.29328.016

The following figure supplement is available for figure 5:

**Figure supplement 1.** Characterization of gatastatin treatment, CAMSAP1 knockout, and KANSL1 knockout.

*Figure 5 continued on next page*

*Figure 5 continued*

DOI: https://doi.org/10.7554/eLife.29328.017

*Figures 1* and *2*, we hypothesized that NuMA localizes dynein activity by recruiting dynactin to minus-ends. Indeed, after NuMA knockout, dynactin (GFP-Arp1A) was no longer detectable at k-fiber minus-ends created by laser ablation (*Figure 4A*; *Video 6*). Immunofluorescence experiments confirmed that in the absence of NuMA, dynactin remained on the spindle but no longer localized to minus-ends (*Figure 4B*). Interestingly, dynein's localization within the spindle (labeled using an antibody against dynein intermediate chain) was less minus-end-specific than dynactin's, both before and after NuMA deletion (*Figure 4C*).

In addition, nocodazole washout experiments revealed that dynactin (p150) localization to individual microtubule minus-ends was lost upon NuMA deletion (*Figure 4D*). Instead, dynactin frequently co-localized with EB1 at plus-ends, similar to its interphase and pre-NEB localization pattern (*Figure 4D*; *Figure 1B*). Thus, the data indicate that NuMA is required for the transport of minus-ends by dynein because NuMA localizes to minus-ends and recruits dynactin there.

## NuMA localizes to minus-ends independently of known minus-end binders

How is NuMA targeted to minus-ends, independently of dynein? In vitro, canonical microtubule-binding regions of NuMA have shown no preference for minus-ends relative to the lattice or plus-end of purified microtubules (*Du et al., 2002*; *Forth et al., 2014*; *Haren and Merdes, 2002*; *Seldin et al., 2016*). Given this lack of minus-end-specific binding in vitro, we hypothesized that NuMA is indirectly recruited to spindle minus-ends by one of three known direct minus-end binders active at mitosis: γ-TuRC (*Zheng et al., 1995*), CAMSAP1 (*Hendershott and Vale, 2014*; *Jiang et al., 2014*) (*Figure 1*; CAMSAP2 and 3 are phosphorylated at mitosis and no longer interact with microtubules [*Jiang et al., 2014*]), and KANSL1/3 (*Meunier et al., 2015*). To test this hypothesis, we treated RPE1 cells with 30 µM gatastatin to block γ-TuRC binding (*Chinen et al., 2015*) (*Figure 5A*; *Figure 5—figure supplement 1A,B*). We also made inducible CRISPR/Cas9 RPE1 cell lines to knock out CAMSAP1 or KANSL1 (*Figure 5—figure supplement 1C,D*), as KANSL1 depletion has been shown to delocalize the entire KANSL complex (*Meunier et al., 2015*). CAMSAP1 knockout caused a small reduction in spindle length (*Figure 5—figure supplement 1E*), consistent with the spindle minus-end protecting function ascribed to its *Drosophila* homolog, Patronin (*Goodwin and Vale, 2010*). To our surprise, however, none of these perturbations qualitatively altered NuMA localization at spindle poles (*Figure 5A*). To check for a more subtle contribution of γ-TuRC, CAMSAP1, or KANSL1 to NuMA localization at minus-ends, we performed k-fiber ablations after 30 µM gatastatin treatment, CAMSAP1 knockout, or KANSL1 knockout and quantified GFP-NuMA recruitment kinetics at new minus-ends. NuMA recruitment to new minus-ends remained robust, and recruitment timescales were statistically indistinguishable from control (*Figure 5B–C*). Thus, the data indicate that the direct mitotic minus-end binders γ-TuRC, CAMSAP1, and KANSL1/3 are not responsible for NuMA's localization to spindle microtubule minus-ends.

Given this lack of involvement of direct minus-end binding proteins, we sought to confirm using super-resolution microscopy that NuMA specifically localizes at individual spindle microtubule minus-ends. Three-dimensional stochastic optical reconstruction microscopy (3D STORM [*Huang et al., 2008*; *Rust et al., 2006*]) of NuMA at k-fiber minus-ends created by ablation revealed organized clusters of NuMA puncta, rather than a lawn of molecules along the lattice near the minus-end (*Figure 5D–G*). The spacing between puncta was consistent with the distance between individual microtubules within k-fibers (*Figure 5H*) as measured by electron microscopy in the same cell type (PtK2) (*McDonald et al., 1992*), supporting the idea that these NuMA puncta are built on individual microtubule minus-ends. A yet undiscovered minus-end binding protein may recruit NuMA; alternatively, NuMA may have direct minus-end-specific binding ability that has not been recapitulated in vitro.

## NuMA function requires both minus-end-recognition and dynactin-recruitment modules

To define the NuMA domain required for spindle minus-end localization, we performed rescue experiments with different NuMA truncations in the NuMA-knockout background (*Figure 6A*). This avoids the C-terminus-mediated oligomerization (*Harborth et al., 1999*) with endogenous protein that can complicate interpretations of localization. In this NuMA-knockout background, full-length NuMA ('FL') localized to spindle minus-ends, as did a bonsai construct ('N-C') with most of the central coiled-coil removed (*Figure 6B*). To our surprise, 'N-C' rescued spindle architecture as effectively as full-length NuMA, indicating that the extraordinary length of NuMA protein is not essential to its function in spindle structure (*Figure 6C*). Importantly, NuMA's C-terminus ('C') alone localized to minus-ends even in the absence of endogenous full-length protein (*Figure 6B*). Its intensity at minus-ends was less striking than that of full-length or N-C protein at poles, likely because minus-ends are not as densely concentrated in disorganized spindles, and perhaps because NuMA's N-terminus facilitates formation of higher order NuMA assemblies. However, the preference of NuMA 'C' for spindle ends was clear. Because dynein-dynactin interacts with NuMA's N-terminus (*Kotak et al., 2012*), the minus-end localization of NuMA's C-terminus provides further support for dynein-independent minus-end recognition.

We sought to more closely define which sections within NuMA's C-terminus (a.a. 1701–2115) are involved in microtubule minus-end localization. The second half of the C-terminus ('C-tail2', a.a. 1882–2115), which contains residues previously implicated in NuMA-microtubule interactions (*Chang et al., 2017*; *Du et al., 2002*; *Gallini et al., 2016*; *Haren and Merdes, 2002*), localized all along spindle microtubules with no minus-end preference (*Figure 6B*). Similarly, C-tail2A (a.a. 1882–1981) bound along the lattice, while C-tail1 (a.a. 1701–1881) did not bind microtubules (*Figure 6—figure supplement 1*). However, in combination ('C-tail1+2A', a.a. 1701–1981) they localized at minus-ends (*Figure 6B*). Indeed, the C-tail1+2A region was sufficient for recruitment to new minus-ends created by ablation, even in the absence of endogenous NuMA (*Figure 6D*; *Video 7*). Which pieces of the tail1+2A region are specifically required for minus-end localization, and what they each do, will require further dissection. Interestingly, related NuMA residues were recently shown to bind at both plus- and minus-ends and play a role in spindle orientation (*Seldin et al., 2016*). In sum, our data indicate that NuMA's localization to spindle microtubule minus-ends is independent of dynein, independent of known direct minus-end binding proteins, and mediated by its C-terminal residues 1701–1981 ('C-tail1+2A').

Importantly, NuMA's C-terminus ('C') localized to minus-ends but could not rescue proper pole focusing or spindle architecture, unlike full-length protein or N-C (*Figure 6C*). This suggests that NuMA's function in spindle organization requires both minus-end binding (via its C-terminus) and the ability to recruit dynactin to minus-ends (via its N-terminus) (*Kotak et al., 2012*). Consistent with this hypothesis, NuMA's C-terminus ('C') was unable to recruit dynactin to minus-ends, while its N-terminus and C-terminus fused ('N-C') did (*Figure 6E*; *Table 1*).

To test whether recruiting dynein-dynactin to the sides of microtubules was sufficient for proper microtubule clustering into poles, we fused NuMA's N-terminus to the Tau microtubule-binding

**Table 1.** NuMA's pole-focusing ability correlates with dynactin- and minus-end-binding.
See also *Figure 6* and *Figure 6—figure supplement 1*.

| Knockout and rescue with NuMA version: | Recruits dynactin | Localizes to lattice (no end preference) | Localizes to preferentially to minus-ends | Rescues pole focusing |
|---|---|---|---|---|
| FL | + | - | + | + |
| N-C | + | - | + | + |
| N-Tau | + | + | - | - |
| C | - | - | + | - |
| C-tail1+2A | *n.d.* | - | + | - |
| C-tail2 | *n.d.* | + | - | - |

*n.d.*=not done, but unlikely to associate with dynactin given that the full C-terminus does not.

DOI: https://doi.org/10.7554/eLife.29328.021

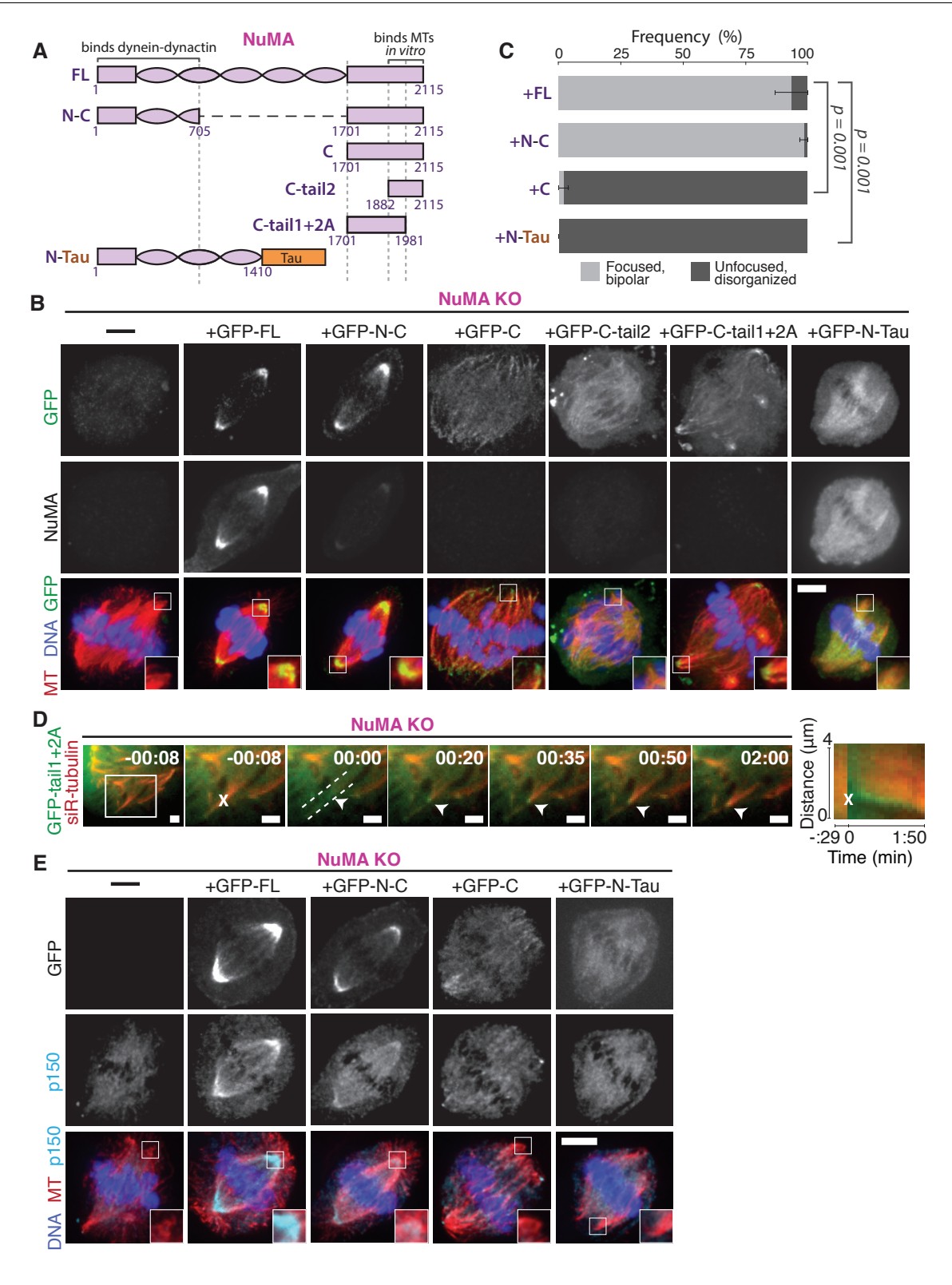

**Figure 6.** NuMA function requires both minus-end-recognition and dynactin-recruitment modules. See also *Table 1*, *Figure 6—figure supplement 1*, and *Video 7*. (**A**) Schematic maps of NuMA truncations and chimeras used. 'FL' indicates full-length NuMA. 'N-C,' 'C,' 'C-tail1,' and 'C-tail2' are NuMA truncations as indicated. 'N-Tau' includes the microtubule binding domain of Tau (orange). Dynein-dynactin bind in the first 705 amino acids of NuMA (*Kotak et al., 2012*). 'C-tail2' was previously implicated in microtubule (MT) binding (*Chang et al., 2017*; *Du et al., 2002*; *Gallini et al., 2016*;

*Figure 6 continued on next page*

*Figure 6 continued*
**Haren and Merdes, 2002**). (B) Representative immunofluorescence images showing localization of GFP-tagged NuMA truncations and chimeras expressed in RPE1 cells in which endogenous NuMA has been knocked out. Canonical microtubule binding domains in tail2 localize all along spindle microtubules, while the addition of tail1 to tail2A confers minus-end localization. Note that NuMA antibody does not recognize NuMA's C-terminus and only weakly recognizes its N-terminus. Scale bar, 5 μm. (C) Graph of the mean percentage of cells ± SEM for each condition from data in (B) that display bipolar spindles with focused poles (light gray) compared to disorganized spindle architecture characteristic of NuMA knockout (detached centrosomes, loss of two focused poles; dark gray). n = 19 ('FL'); n = 32 ('N-C'); n = 20 ('C'); n = 16 ('N-Tau') from three to five independent experiments. p-Values calculated by Tukey post-hoc test after one-way ANOVA ($F$(3,10) = 267, p<0.00001). (D) Representative time-lapse live images and kymograph of RPE1 cell expressing GFP-C-tail1+2A in a NuMA-knockout background. After ablation ('X'), C-tail1+2A is recruited to k-fiber minus-ends (arrowhead). The k-fiber stub slowly polymerizes, but its minus-end is not transported by dynein and remains detached from the spindle. Time is in min:s, with the frame captured immediately following ablation set to 00:00 s. Scale bars, 2 μm. Kymograph (right) taken along dashed line path. (E) Representative immunofluorescence images showing localization of dynactin (p150) after NuMA knockout and rescue with GFP-tagged NuMA truncations and chimeras in RPE1 cells. Constructs containing NuMA's N-terminus recruit p150. Scale bar, 5 μm.
DOI: https://doi.org/10.7554/eLife.29328.018
The following figure supplement is available for figure 6:

**Figure supplement 1.** NuMA 'C-tail1' and 'C-tail2A' alone do not localize to minus-ends.
DOI: https://doi.org/10.7554/eLife.29328.019

domain ('N-Tau'). N-Tau localized along the length of spindle microtubules and recruited dynactin there (*Figure 6E*). Despite combining dynein-dynactin binding and microtubule lattice binding, N-Tau was not enriched at minus-ends and was unable to rescue spindle architecture (*Figure 6C*; *Table 1*).

Like a traditional cargo adaptor, NuMA may target force to spindle minus-ends using a cargo (minus-end) binding module ('C') and a dynactin-recruitment module ('N'). Furthermore, the inability of N-Tau to rescue spindle architecture in the absence of endogenous NuMA suggests that specifically targeting dynein-dynactin to minus-ends, not just all along spindle microtubules as N-Tau does, is critical for organizing a focused, bipolar spindle.

## Discussion

### NuMA targets dynactin to minus-ends, spatially regulating dynein activity at mitosis

Our data indicate that NuMA localizes to spindle microtubule minus-ends independently of dynein and minus-end binding proteins γ-TuRC, CAMSAP1, and KANSL1/3. NuMA then targets dynactin to minus-ends, localizing dynein motor activity there. In addition, targeting dynein to the end of its track could permit amplification by motor pile-up, as NuMA at minus-ends captures processive dynein complexes. Altogether, our findings are consistent with a model in which NuMA confers minus-end targeting of the dynein-dynactin complex upon nuclear envelope breakdown (NEB), when NuMA is released from the nucleus. Indeed, active microtubule clustering by dynein is first observed coincident with NEB (*Rusan et al., 2002*). Thus, NuMA may provide both spatial (minus-end-specific) and temporal (mitosis-specific) regulation of dynein-powered force (*Figure 7A*).

The data also indicate that NuMA's minus-end localization at mitosis is mediated by the tail1+2A region of its C-terminus, while previously identified microtubule-binding domains tail2A or tail2B (*Du et al., 2002*; *Gallini et al., 2016*; *Haren and Merdes, 2002*) are not sufficient. We propose three hypotheses for tail1+2A-mediated minus-end recognition. First, a yet-unidentified minus-end binding protein or the minus-end-directed kinesin HSET (*Gaglio et al., 1996*; *Mountain et al., 1999*) could localize NuMA via an interaction that requires this longer segment of the NuMA tail. During the preparation of this manuscript, ASPM was identified as a novel mammalian spindle minus-end binder, but ASPM deletion does not affect NuMA localization (*Jiang et al., 2017*). Second, NuMA may recognize both plus- and minus-ends via microtubule curvature sensing, as previously proposed (*Seldin et al., 2016*), and yet be actively excluded from plus-ends within the spindle by other proteins. Third, NuMA tail1+2A may bind minus-ends directly in cells, perhaps using post-translational modifications (*Compton and Luo, 1995*; *Gallini et al., 2016*; *Magescas et al., 2017*; *Yan et al., 2015*) not previously recapitulated in vitro.

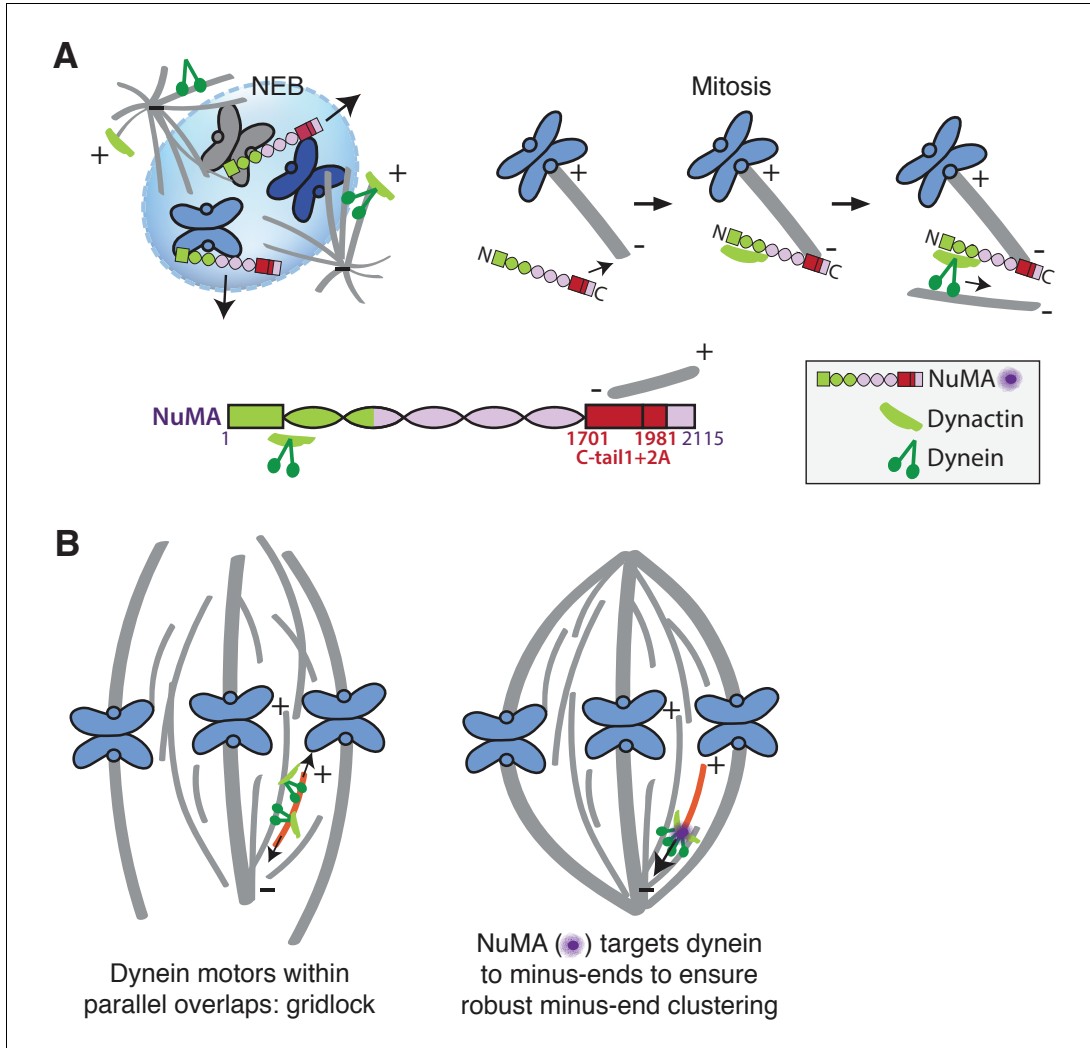

**Figure 7.** Model: NuMA spatially targets dynein activity to minus-ends at mitosis to ensure minus-end clustering into poles. (**A**) Our findings suggest that NuMA confers minus-end targeting to the dynein-dynactin complex upon NEB, when NuMA is released from the nucleus. NuMA localization at minus-ends is mediated by a region of its C-terminus (amino acids 1701–1981, in red) that includes tail1 in addition to lattice-binding domain tail2A. NuMA targets dynactin to minus-ends, localizing dynein motor activity. (**B**) Model: Without minus-end targeting, dynein molecules acting on the red microtubule oppose each other, resulting in gridlock (left). NuMA-mediated targeting of force to minus-ends allows for productive clustering of the red microtubule into the pole (right). Altogether, the data support a model for mammalian spindle organization in which targeting poleward force to microtubule minus-ends specifically – by NuMA-mediated dynactin recruitment – provides robust clustering of microtubules into a focused, bipolar spindle.

DOI: https://doi.org/10.7554/eLife.29328.022

## Function of force at spindle microtubule minus-ends

Models of spindle assembly in silico predict that enriching a minus-end-directed motor at microtubule ends can break gridlock between parallel microtubules and allow robust minus-end clustering into poles (*Burbank et al., 2007*; *Goshima et al., 2005*; *Hyman and Karsenti, 1996*; *Surrey et al., 2001*). However, a lack of mechanistic information and tools made it difficult to test such hypotheses in vivo. The present study reveals a mechanism for direct recruitment of the motor to minus-ends, via NuMA, and is consistent with the prediction that targeting dynein-dynactin to the sides of microtubules is not sufficient for robust spindle organization. Both the minus-end-binding module ('C') and dynein-dynactin-binding module ('N') of NuMA are required for bipolar focusing, while the

distance between them is not critical for function. Fusing the dynein-dynactin-binding module to the Tau microtubule binding domain is not sufficient, suggesting a requirement for minus-end-specific forces (*Figure 6*; *Table 1*). However, we cannot formally exclude that features of the NuMA C-terminus (missing from 'N-Tau') other than minus-end binding enable rescue of pole focusing, or that while N-Tau recruits dynactin (*Figure 6D*), it cannot effectively couple to dynein. Altogether, the work is consistent with a model for mammalian spindle organization in which targeting poleward force to microtubule minus-ends (by NuMA-mediated dynactin recruitment) is critical for organizing microtubules into a focused, bipolar array (*Figure 7B*). More broadly, the emergence of pole architecture at mitosis illustrates how spatial regulation of molecular-scale activities (like NuMA localizing dynein activity at individual minus-ends) can give rise to complex and diverse cellular-scale structures.

### NuMA as mitotic dynein adaptor or activator

The data suggest that NuMA in the mammalian spindle body may function as a traditional cargo adaptor, with a cargo (minus-end)-binding module and a motor-binding module. In the spindle body context, microtubule minus-ends are the cargo, just as the cortex can be framed as cargo for cortical dynein – analogous to more canonical interphase cargo like membranous vesicles or organelles. This framework raises the question of whether NuMA additionally serves as a dynein activator, inducing highly processive motility like interphase dynein adaptors can (*McKenney et al., 2014*; *Schlager et al., 2014*).

Dynein's localization in mammalian mitosis is more ubiquitous than that of dynactin (*Figure 4C*; *Figure 2—figure supplement 1C*); high cytoplasmic levels of dynein prevented us from detecting clear dynein enrichment at ablation-created minus-ends, for example (data not shown), and dynein localization is not as altered by NuMA knockout as dynactin localization (*Figure 4C*). These observations could be explained if NuMA and dynactin at minus-ends selectively activate dynein there – through a conformational shift that renders the motor more processive, for example – rather than simply selectively localizing it there. In other words, dynein's localization within the spindle body may be less tightly regulated than its activity. The loss of dynein activity at minus-ends observed after NuMA knockout (*Figure 3C*) may stem from a lack of dynein activation (without NuMA and dynactin present at minus-ends) rather than a lack of dynein enrichment.

Unlike known dynein activators, NuMA is thought to not only homodimerize but also oligomerize into higher order assemblies (*Harborth et al., 1999*; *Saredi et al., 1997*). NuMA's oligomerization ability suggests that it could assemble teams of motors and invites a comparison to motor-clustering in microdomains on large cellular cargoes, like phagosomes (*Rai et al., 2016*). The increased force and processivity provided by teams of NuMA-dynactin-dynein complexes on mitotic minus-ends could enable transport and clustering of minus-ends despite high loads and friction created by dense microtubule crosslinking and – in the case of k-fiber minus-ends – coupling to chromosomes.

## Materials and methods

**Key resources table**

| Reagent type (species) or resource | Designation | Source or reference | Identifiers | Additional information |
|---|---|---|---|---|
| Cell line (*Potorous tridactylus*) (male) | PtK2 | T. Mitchison | ATCC Cat#CCL-56; RRID: CVCL_0514 | kidney epithelial |
| Cell line (*P. tridactylus*) (male) | PtK2 GFP-tubulin | PMID: 12604591 | | kidney epithelial, stably expressing GFP-α-tubulin |
| Cell line (*Homo sapiens*) (female) | RPE1 | ATCC | ATCC Cat#CRL-4000; RRID: CVCL_4388 | retina, epithelial |
| Cell line (*H. sapiens*) (female) | RPE1 NuMA knockout | this paper | | RPE1 with stably integrated spCas9 (Tet-On promoter) and NuMA sgRNA #2 |
| Cell line (*H. sapiens*) (female) | RPE1 CAMSAP1 knockout | this paper | | RPE1 with stably integrated spCas9 (Tet-On promoter) and CAMSAP1 sgRNA #1 |

*Continued on next page*

*Continued*

| Reagent type (species) or resource | Designation | Source or reference | Identifiers | Additional information |
|---|---|---|---|---|
| Cell line (*H. sapiens*) (female) | RPE1 KANSL1 knockout | this paper | | RPE1 with stably integrated spCas9 (Tet-On promoter) and KANSL1 sgRNA #3 |
| Cell line (*H. sapiens*) (female) | HeLa dynein heavy chain (DHC) knockout | PMID: 28216383 | cmk1a DYNC1H1 sgD1.1 | RPE1 with stably integrated spCas9 (Tet-On promoter) and DHC sgRNA |
| Recombinant DNA reagent (plasmid) | GFP-Arp1A | I. Cheeseman | Addgene 4432 | Progenitor: pBABE variant |
| Recombinant DNA reagent (plasmid) | 2xGFP-Arp1A | this paper | | Progenitor: GFP-Arp1A |
| Recombinant DNA reagent (plasmid) | DsRed-p150$^{217-548}$ (CC1) | PMID: 12391026 | | Progenitor: pDsRed-N1 (Clontech) |
| Recombinant DNA reagent (plasmid) | mCherry-p50 | PMID: 19196984 | | Progenitor: mCherry-C1 (Clontech) |
| Recombinant DNA reagent (plasmid) | GFP-CAMSAP1 | PMID: 24486153 | | Progenitor: pEGFP-C1 (Clontech) |
| Recombinant DNA reagent (plasmid) | GFP-NuMA | PMID: 15561764 | | Progenitor: pEGFP-N1 (Clontech) |
| Recombinant DNA reagent (plasmid) | GFP-NuMA_resistant | this paper | | Progenitor: GFP-NuMA. Invisible to NuMA sgRNA #2. NM_006185.3 |
| Recombinant DNA reagent (plasmid) | GFP-NuMA 'N-C' | this paper | | Progenitor: GFP-NuMA_resistant |
| Recombinant DNA reagent (plasmid) | GFP-NuMA 'C' | this paper | | Progenitor: GFP-NuMA_resistant |
| Recombinant DNA reagent (plasmid) | GFP-NuMA 'C-tail1' | this paper | | Progenitor: GFP-NuMA_resistant |
| Recombinant DNA reagent (plasmid) | GFP-NuMA 'C-tail2' | this paper | | Progenitor: GFP-NuMA_resistant |
| Recombinant DNA reagent (plasmid) | GFP-NuMA 'C-tail2A' | this paper | | Progenitor: GFP-NuMA_resistant |
| Recombinant DNA reagent (plasmid) | GFP-NuMA 'C-tail2B' | this paper | | Progenitor: GFP-NuMA_resistant |
| Recombinant DNA reagent (plasmid) | GFP-N-Tau | this paper | | Progenitors: GFP-NuMA_resistant; pmEmerald-MAPTau-C10 (M.Davidson) |
| Antibody | anti-α-tubulin (DM1α; mouse) | Sigma | Sigma T6199 | IF (1:1000); WB (1:5000). RRID: AB_477583 |
| Antibody | anti-α-tubulin (DM1α; mouse) conjugated to AF488 | Cell Signaling | Cell Signaling 8058S | IF (1:200). RRID: AB_10860077 |
| Antibody | anti-α-tubulin (rabbit) | Abcam | Abcam ab18251 | IF (1:500). RRID: AB_2210057 |
| Antibody | anti-NuMA (rabbit) | Novus Biologicals | Novus Biologicals NB500-174 | IF (1:400); WB (1:1000). RRID: AB_10002562 |
| Antibody | anti-p150-Glued (mouse) | BD Biosciences | BD Biosciences 610473 | IF (1:400). RRID: AB_397845 |
| Antibody | anti-dynein intermediate chain (mouse) | Millipore | Millipore MAB1618MI | IF (1:250); WB (1:250). RRID: AB_2246059 |
| Antibody | anti-EB1 (rabbit) | Santa Cruz Biotech | Santa Cruz sc-15347 | IF (1:100). RRID: AB_2141629 |
| Antibody | anti-γ-tubulin (rabbit) | Sigma | Sigma T3559 | IF (1:500). RRID: AB_477575 |
| Antibody | anti-KANSL1 (rabbit) | Abnova | Abnova PAB20355 | WB (1:500). RRID: AB_10984400 |
| Antibody | anti-CAMSAP1 (rabbit) | Novus Biologicals | Novus Biologicals NBP1-26645 | WB (1:500). RRID: AB_1852845 |
| Antibody | anti-GFP (camel) conjugated to Atto488 | ChromoTek | ChromoTek gba-488 | IF (1:500). RRID: AB_2631434 |

*Continued on next page*

*Continued*

| Reagent type (species) or resource | Designation | Source or reference | Identifiers | Additional information |
|---|---|---|---|---|
| Dye | siR-tubulin | Cytoskeleton, Inc. | Cytoskeleton Inc. CYSC-002 | 100 nM |
| Drug | verapamil | Cytoskeleton, Inc. | Cytoskeleton Inc. CYSC-002 | 10 µM |
| Drug | gatastatin | PMID: 26503935 | | 30 µM |
| Drug | nocodazole | Sigma | Sigma M1404 | 5 µM |
| Drug | STLC | Sigma | Sigma 164739 | 10 µM |
| Sequence-based reagent | NuMA sgRNA #1 | this paper | | 5'-ATGACACTCCACGCCACCCG-3' |
| Sequence-based reagent | NuMA sgRNA #2* | this paper | | 5'-AAGTCCAGTCTCTCTGACAC-3' |
| Sequence-based reagent | NuMA sgRNA #3 | this paper | | 5'-ACAGCAAATCTTGAAGCAGC-3' |
| Sequence-based reagent | CAMSAP1 sgRNA #1* | this paper | | 5'-GCCGCGTCGTAGCGGTCCAG-3' |
| Sequence-based reagent | CAMSAP1 sgRNA #2 | this paper | | 5'-CCGACAGTCTGTATAATATT-3' |
| Sequence-based reagent | CAMSAP1 sgRNA #3 | this paper | | 5'CCGAATATTATACAGACTGT-3' |
| Sequence-based reagent | KANSL1 sgRNA #1 | this paper | | 5'-GAGCCAGTTTGAACCGGATA-3' |
| Sequence-based reagent | KANSL1 sgRNA #2 | this paper | | 5'-ACACCATATCCGGTTCAAAC-3' |
| Sequence-based reagent | KANSL1 sgRNA #3* | this paper | | 5'-GAGCCCGTTTTCCCCCATTG-3' |

*Guide RNA expressed by cell line used for subsequent experiments, following initial verification of consistent spindle phenotypes.

## Cell culture and transfection

PtK2 cells were cultured in MEM (11095; Thermo Fisher, Waltham, MA) supplemented with sodium pyruvate (11360; Thermo Fisher), nonessential amino acids (11140; Thermo Fisher), penicillin/streptomycin, and 10% heat-inactivated fetal bovine serum (FBS) (10438; Thermo Fisher). RPE1 and HeLa cells were cultured in DMEM/F12 with GlutaMAX (10565018; Thermo Fisher) supplemented with penicillin/streptomycin and 10% FBS. For Tet-on inducible CRISPR-Cas9 cell lines, tetracycline-screened FBS (SH30070.03T; Hyclone Labs, Logan, UT) was used. Cell lines were not STR-profiled for authentication. All cell lines tested negative for mycoplasma. Cells were maintained at 37°C and 5% $CO_2$ and were transfected with DNA using ViaFect (E4981; Promega, Madison, WI) 48 hr (RPE1/HeLa) or 72 hr (PtK2) before imaging.

## Inducible CRISPR-Cas9 knockout cells

sgRNAs were designed against 5' exons of NuMA, CAMSAP1, and KANSL1 using http://crispr.mit.edu. sgRNAs are listed in Key Resource Table. The plasmid used to express sgRNAs under control of the hU6 promoter (pLenti-sgRNA) was a gift from T. Wang, D. Sabatini, and E. Lander (Whitehead/Broad/MIT). An RPE1 cell line containing doxycycline-inducible human codon-optimized spCas9 was a gift from I. Cheeseman (Whitehead/MIT) and was generated as described in (*McKinley et al., 2015*) using a derivative of the transposon described in (*Wang et al., 2014*). We infected this inducible-spCas9 RPE1 cell line with each pLenti-sgRNA as described in (*Wang et al., 2015*) using virus expressed in HEK293T cells and 10 µg/mL polybrene and selected with 6 µg/mL puromycin. For each targeted gene, we tested 3 independent sgRNA sequences, each of which generated indistinguishable spindle phenotypes (data not shown), and picked one line for subsequent studies. Four days before each experiment, spCas9 expression was induced with 1 µM doxycycline hyclate.

## Live imaging and laser ablation

For live imaging, cells were plated on glass-bottom 35 mm dishes coated with poly-D-lysine (MatTek Corporation, Ashland, MA) and imaged in a stage-top humidified incubation chamber (Tokai Hit, Fujinomiya-shi, Japan) maintained at 30°C and 5% CO2. To visualize tubulin, 100 nM siR-Tubulin dye (Cytoskeleton, Inc., Denver, CO) was added 2 hr prior to imaging, along with 10 µM verapamil (Cytoskeleton, Inc.). Under these conditions, there was no detected defect in spindle appearance or microtubule dynamics. As described elsewhere (*Elting et al., 2014*), cells were imaged using a spinning disk confocal inverted microscope (Eclipse Ti-E; Nikon Instruments, Melville, NY) with a 100 × 1.45 Ph3 oil objective through a 1.5X lens, operated by MetaMorph (7.7.8.0; Molecular Devices, Sunnyvale, CA). Laser ablation (30 3-ns pulses at 20 Hz) with 551 nm light was performed using the galvo-controlled MicroPoint Laser System (Andor, Belfast, UK). For laser ablation experiments, images were acquired more slowly prior to ablation and more rapidly after ablation (typically 7 s prior and 3.5 s after ablation).

## Immunofluorescence and antibodies

For immunofluorescence, cells were plated on #1.5 25 mm coverslips coated with 1 mg/mL poly-L-lysine. Cells were fixed with 95% methanol + 5 mM EGTA at −20°C for 3 min, washed with TBS-T (0.1% Triton-X-100 in TBS), and blocked with 2% BSA in TBS-T for 1 hr. Primary and secondary antibodies were diluted in TBS-T+2% BSA and incubated with cells overnight at 4°C (primary) or for 20 min at room temperature (secondary). DNA was labeled with Hoescht 33342 (Sigma, St. Louis, MO) before cells were mounted in ProLongGold Antifade (P36934; Thermo Fisher). Cells were imaged using the spinning disk confocal microscope described above. Antibodies: mouse anti-α-tubulin DM1α (T6199; Sigma), rabbit anti-α-tubulin (ab18251; Abcam, Cambridge, UK), rabbit anti-NuMA (NB500-174; Novus Biologicals, Littleton, CO), mouse anti-p150-Glued (610473; BD Biosciences, San Jose, CA), mouse anti-α-tubulin DM1α conjugated to Alexa488 (8058S; Cell Signaling, Danvers, MA), mouse anti-dynein intermediate chain (MAB1618MI; Millipore, Billerica, MA), rabbit anti-EB1 (sc-15347; Santa Cruz Biotechnology, Dallas, TX), rabbit anti-KANSL1 (PAB20355; Abnova, Taipei City, Taiwan), rabbit anti-CAMSAP1 (NBP1-26645; Novus Biologicals), mouse anti-actin (MAB1501; Millipore), rabbit anti-γ-tubulin (T3559; Sigma), and camel nanobody against GFP coupled to Atto488 (gba-488; ChromoTek, Hauppauge, NY).

## STORM

PtK2 cells expressing GFP-α-tubulin (gift of A. Khodjakov, Wadsworth Center) were plated on photo-etched, gridded coverslips (G490; ProSciTech, Kirwan, Australia) coated with 1 mg/mL poly-L-lysine (P-1524; Sigma) and imaged at 29–30°C in a homemade heated aluminum coverslip holder using the confocal microscope and ablation system described above. 20–30 s after k-fiber ablation, imaging media was replaced with fixative (as above) chilled to −80°C, and the coverslip holder was placed on ice for 1 min. Cells were incubated with 3% BSA in PBS for 1 hr at room temperature, and then with primary antibodies overnight at 4°C. Secondary antibodies ((anti-mouse Cy3B; Jackson Immunoresearch, West Grove, PA); anti-rabbit AF647 (Life Tech, Carlsbad,CA)) were incubated for 30 min at room temperature. Antibody incubations were followed by four washes with 0.2% BSA in PBS. Samples were stored in PBS during confocal imaging, and coverslip grid was used to re-find the individual ablated cell. For 3D STORM imaging, samples were mounted on glass slides with a standard STORM imaging buffer consisting of 5% (w/v) glucose, 100 mM cysteamine, 0.8 mg/mL glucose oxidase, and 40 µg/mL catalase in 1M Tris-HCl (pH 7.5) (*Huang et al., 2008*; *Rust et al., 2006*). Coverslips were sealed using Cytoseal 60. STORM imaging was performed on a homebuilt setup based on a modified Nikon Eclipse Ti-E inverted fluorescence microscope using a Nikon CFI Plan Apo λ 100x oil immersion objective (NA 1.45). Dye molecules were photoswitched to the dark state and imaged using either 647- or 560 nm lasers (MPB Communications, Montreal, CAN); these lasers were passed through an acousto-optic tunable filter and introduced through an optical fiber into the back focal plane of the microscope and onto the sample at intensities of ~2 kW cm$^{-2}$. A translation stage was used to shift the laser beams towards the edge of the objective so that light reached the sample at incident angles slightly smaller than the critical angle of the glass-water interface. A 405 nm laser was used concurrently with either the 647- or 560 nm lasers to reactivate fluorophores into

the emitting state. The power of the 405 nm laser (typical range 0–1 W cm$^{-2}$) was adjusted during image acquisition so that at any given instant, only a small, optically resolvable fraction of the fluorophores in the sample were in the emitting state. Emission was recorded with an Andor iXon Ultra 897 EM-CCD camera at a framerate of 220 Hz, for a total of ~120,000 frames per image. For 3D STORM imaging, a cylindrical lens of focal length 1 m was inserted into the imaging path so that images of single molecules were elongated in opposite directions for molecules on the proximal and distal sides of the focal plane (*Huang et al., 2008*). Two-color imaging was performed via sequential imaging of targets labeled by AF647 and Cy3B. The raw STORM data was analyzed according to previously described methods (*Huang et al., 2008*; *Rust et al., 2006*).

## Drug treatment and microtubule re-growth

To inhibit γ-tubulin, 30 µM gatastatin (gift of Takeo Usui and Ichiro Hayakawa, University of Tsukuba and Okayama University, respectively) (*Chinen et al., 2015*) was added 25–60 min before imaging (*Figure 5A–C*, *Figure 5—figure supplement 1A*). To test microtubule re-growth in the presence of gatastatin (*Figure 5—figure supplement 1B*), all cells were treated with 10 µM STLC for 6 hr to create monopolar spindles. For cold treatment, cells were then placed on ice; after 1 hr, 30 µM gatastatin or equivalent (0.1%) DMSO was added. After 10 more minutes on ice, cells were moved to room temperature for 1 min before fixation. Control cells (no ice) were incubated in media containing 30 µM gatastatin or DMSO for 11 min at room temperature before fixation. For all microtubule re-growth experiments after nocodazole washout, cells were treated with 5 µM nocodazole (M1404; Sigma) for 15 min at 37°C. After three washes, cells were incubated at room temperature for 6–8 min before fixation and immunofluorescence.

## Plasmids

2xGFP-Arp1A was made by inserting EGFP from pEGFP-N1 (Clontech, Takara Bio USA, Mountain View, CA) by Gibson assembly between GFP and Arp1A of GFP-Arp1A (human Arp1A in a pBABE variant, Addgene 4432; gift from I. Cheeseman, Whitehead Institute) (*Kiyomitsu and Cheeseman, 2012*). 2x-GFP-Arp1A localized correctly to kinetochores and poles, and spindle organization was unperturbed. To make Cas9-resistant GFP-NuMA ('GFP-NuMA_resistant'), full-length human NuMA (NM_006185.3) with silent mutations (5'-GTGTCAGAGAGACTGGACTTT-3' mutated to 5'-GTTAGTGAACGCTTGGATTTT-3', preserving amino acids 57–62 of NP_006176.2 ('VSERLD')) was synthesized and cloned (Epoch Life Science, Missouri City, TX) into pEGFP-N1 at BglII and EcoRI sites. Subsequent truncations of NuMA ('N-C', 'C', 'C-tail1', 'C-tail2', 'C-tail2A', 'C-tail2B') were synthesized and cloned (Epoch Life Science) into 'GFP-NuMA_resistant' at BglII and HindIII sites. To make GFP-N-Tau, NuMA amino acids 1–1410 from 'GFP-NuMA_resistant' followed by a flexible linker and MAPTau (NM_01684.1) from pmEmerald-MAPTau-C-10 (gift from M. Davidson, Florida State University) were synthesized and cloned (Epoch Life Science) into 'GFP-NuMA_resistant' at HindIII and SalI sites. Other plasmids used: DsRed-p150$^{217-548}$ (CC1; amino acids 217–548 of chicken p150 in

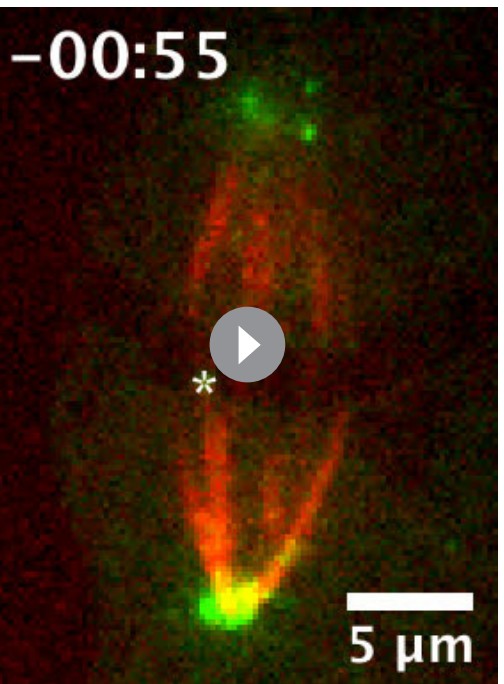

**Video 1.** Dynactin is robustly recruited to new spindle microtubule minus-ends. See also *Figure 1C*. Live confocal imaging of a PtK2 cell expressing GFP-Arp1A, with spindle microtubules labeled by siR-tubulin. GFP-Arp1A is recruited to k-fiber minus-ends (arrowhead) created by ablation ('X') and moves with them as dynein pulls them poleward (*Elting et al., 2014*). Time is in min:s, with the frame captured immediately following ablation set to 00:00. Plus-end of ablated k-fiber is marked by '*'. Scale bar, 5 µm.
DOI: https://doi.org/10.7554/eLife.29328.005

pDsRed-N1, Clontech, gift from T. Schroer, Johns Hopkins University) (*Quintyne and Schroer, 2002*); mCherry-p50 (chicken p50 in mCherry-C1, Clontech, gift from M. Moffert and T. Schroer, Johns Hopkins University) (*Shrum et al., 2009*); GFP-NuMA (human NuMA in pEGFP-N1, Clontech, gift from D. Compton, Dartmouth Medical School) (*Kisurina-Evgenieva et al., 2004*); GFP-CAM-SAP1 (human CAMSAP1 in pEGFP-C1, Clontech, gift from A. Akhmanova, Utrecht University) (*Jiang et al., 2014*).

## Data analysis

To determine the percentage of p150 at plus-ends vs. minus-ends (*Figure 1B*, *Figure 4D*), we used single microtubules where both ends were clearly visible. We found that EB1 consistently labeled just one end, the plus-end. We determined p150 localization relative to the EB1-labeled plus-end and calculated the percentage of p150 at each location within each cell. Percentages for multiple cells were averaged for *Figures 1B* and *4D*. Pre-NEB cells were distinguished from post-NEB cells by the exclusion of microtubules from the nucleus, circle-shaped chromosome packing in the nucleus, and, when possible, NuMA localization within the nucleus.

Kymographs of GFP-Arp1A, GFP-NuMA, and GFP-CAMSAP1 puncta and pole position over time (*Figure 1C–E*, *Figure 2C*, *Figure 6D*) were generated in ImageJ (Version 2.0.0/1.51 hr). To measure GFP intensity at ablation-created minus-ends over time (*Figure 1F*, *Figure 2D*, *Figure 2—figure supplement 1E*, *Figure 5B*), we used a home-written Matlab (R2012a Version 7.4) program to integrate GFP intensity within a 1.4 µm-diameter circle centered on the manually-tracked k-fiber minus-end, and to measure local background intensity within a surrounding 2.7 µm-diameter 'donut'. Code is available at https://github.com/chueschen/IntensityAtMinusEnd (*Hueschen, 2017*; copy

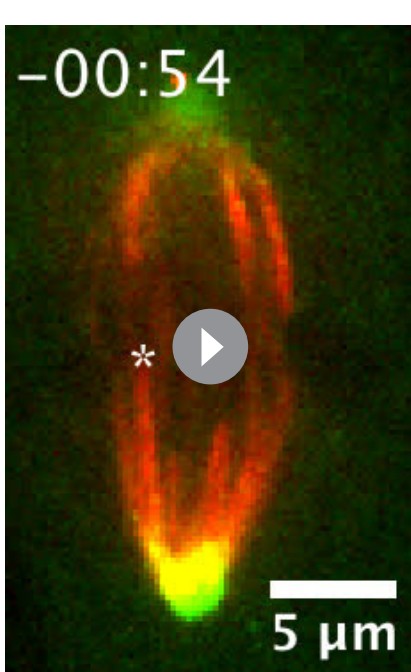

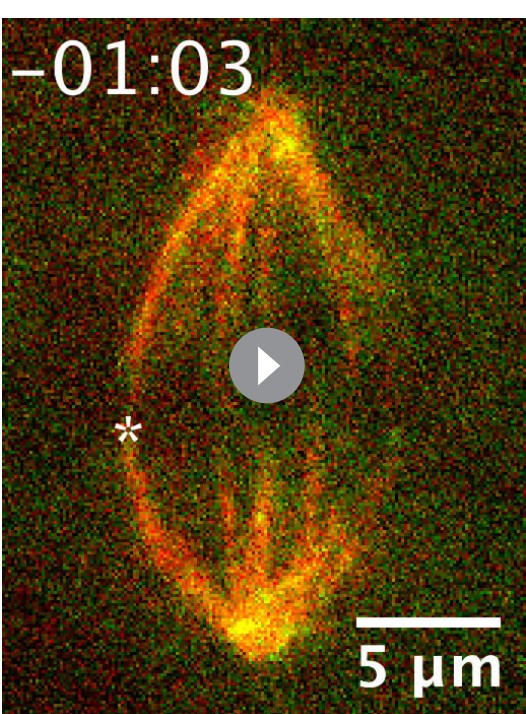

**Video 2.** NuMA is robustly recruited to new spindle microtubule minus-ends. See also *Figure 1D*. Live confocal imaging of a PtK2 cell expressing GFP-NuMA, with spindle microtubules labeled by siR-tubulin. GFP-NuMA is quickly recruited to k-fiber minus-ends (arrowhead) created by ablation ('X') and moves with them as dynein pulls them poleward (*Elting et al., 2014*). Time is in min:s, with the frame captured immediately following ablation set to 00:00. Plus-end of ablated k-fiber is marked by '*'. Scale bar, 5 µm.
DOI: https://doi.org/10.7554/eLife.29328.006

**Video 3.** CAMSAP1 is robustly recruited to new spindle microtubule minus-ends. See also *Figure 1E*. Live confocal imaging of a PtK2 cell expressing GFP-CAMSAP1, with spindle microtubules labeled by siR-tubulin. GFP-CAMSAP1 is immediately recruited to k-fiber minus-ends (arrowhead) created by ablation ('X') and moves with them as dynein pulls them poleward. Time is in min:s, with the frame captured immediately following ablation set to 00:00. Plus-end of ablated k-fiber is marked by '*'. Scale bar, 5 µm.
DOI: https://doi.org/10.7554/eLife.29328.007

 subtraction, the intensity measured at the cut site during the three frames before ablation (k-fiber intensity) was averaged and set to zero. For NuMA and Arp1A, we then fit a sigmoid function ($y = \frac{a}{1+e^{\frac{-(x-b)}{c}}}$ where $y$ = intensity and $x$ = time) to each trace, normalized to plateau height ($a$ = 1), and solved for the time $b$ at which $y = 0.5*a$ to determine time to half-maximum intensity (*Figure 1G*, *Figure 2E*, *Figure 2—figure supplement 1F*, *Figure 5C*). For CAMSAP1, we normalized each trace to peak height (mean intensity from t = 5 s to t = 20 s) and found the first point at which intensity passed 0.5 to determine time to half-maximum intensity. Finally, to generate mean intensity traces, data from all traces were collected into 5 s wide bins in time and their intensities were averaged. Stub length (distance between k-fiber plus- and minus-ends, *Figure 1H*) was measured in ImageJ at the first frame following ablation.

Minus-end position data (*Figure 3C*) were generated by manual tracking of ablation-created k-fiber minus-ends (marked by GFP-CAMSAP1) and spindle poles in time-lapse videos, using a second home-written Matlab program (*Elting et al., 2014*). Nearest neighbor distances between NuMA puncta in STORM imaging (*Figure 5H*) were measured as the center-to-center distance from each NuMA puncta to its nearest neighboring puncta. NuMA truncation rescue capability (*Figure 6C*) reports the percentage of bipolar spindles with two focused poles compared to disorganized spindle architecture characteristic of NuMA knockout (detached centrosomes, loss of k-fiber focusing into two poles). Percentage was calculated for each experiment ($n$ = 3–5 experiments) and then averaged.

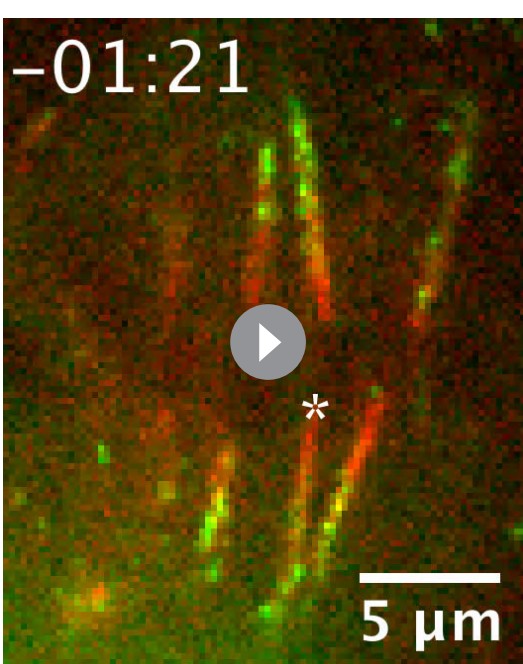

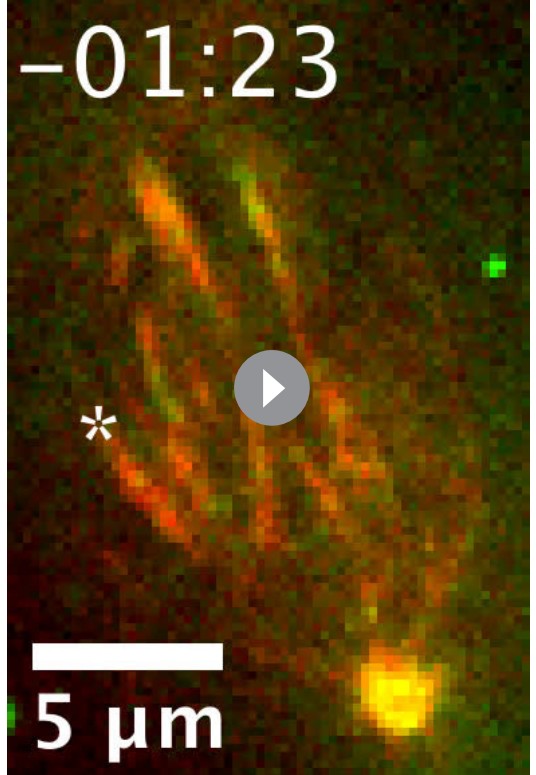

**Video 4.** NuMA quickly localizes to k-fiber minus-ends despite dynein-dynactin inhibition. See also *Figure 2C*. Live confocal imaging of a PtK2 cell expressing GFP-NuMA, in which dynein-dynactin is inhibited by overexpression of p50 (dynamitin). K-fibers are unfocused and splayed due to dynein-dynactin inhibition, but NuMA is still robustly recruited to k-fiber minus-ends (arrowheads) created by ablation ('X'). Time is in min:s, with frame captured immediately following ablation set to 00:00. Plus-end of ablated k-fiber is marked by '*'. Scale bar, 5 μm.
DOI: https://doi.org/10.7554/eLife.29328.010

**Video 5.** NuMA is required for transport of minus-ends by dynein. See also *Figure 3A*. Live confocal imaging of a RPE1 cell in which NuMA has been knocked out using an inducible Cas9 system, and GFP-CAMSAP1 is expressed to label minus-ends. After ablation ('X'), the k-fiber minus-end (arrowhead) is not transported poleward by dynein and remains detached from the disorganized spindle. Time is in min:s, with the frame captured immediately following ablation set to 00:00 s. Plus-end of ablated k-fiber is marked by '*'. Scale bar, 5 μm.
DOI: https://doi.org/10.7554/eLife.29328.013

## Statistics

All data are expressed as average ±standard error of the mean (SEM). Calculations of correlation coefficients (Pearson's *r*) and p values were performed in Matlab. One-way ANOVA and Tukey post-hoc tests (*Figures 1G*, *5C* and *6C*) were performed in Microsoft Excel and Matlab. All other p values were calculated using two-tailed unpaired *t*-tests with GraphPad Software. Quoted *n*'s are described in more detail in Figure Legends, but in general refer to individual biological structures analyzed (biological replicates, for example, individual spindle lengths, individual k-fiber ablations, etc.).

## Image presentation

Time-lapse images (*Figures 1C–E*, *2C*, *3A*, *4A*, *5D* and *6D*) show a single spinning disk confocal slice, as do immunofluorescence images of microtubule re-growth after nocodazole or cold treatment (*Figures 1A–B* and *4D*, *Figure 5—figure supplement 1B*) and post-ablation confocal immunofluorescence images (*Figure 5E*, *Figure 2—figure supplement 1A–B*). 3D STORM images (*Figure 5F–G*) show a single 600 nm slice in Z. Immunofluorescence images of spindles (*Figures 2F*, *4B, C*, *5A*, *6B and E* and *Figure 1—figure supplement 1*, *Figure 2—figure supplement 1D*, *Figure 6—figure supplement 1B*) show max intensity projections (1–2 µm in Z) of spinning disk confocal Z-stacks.

## Video preparation

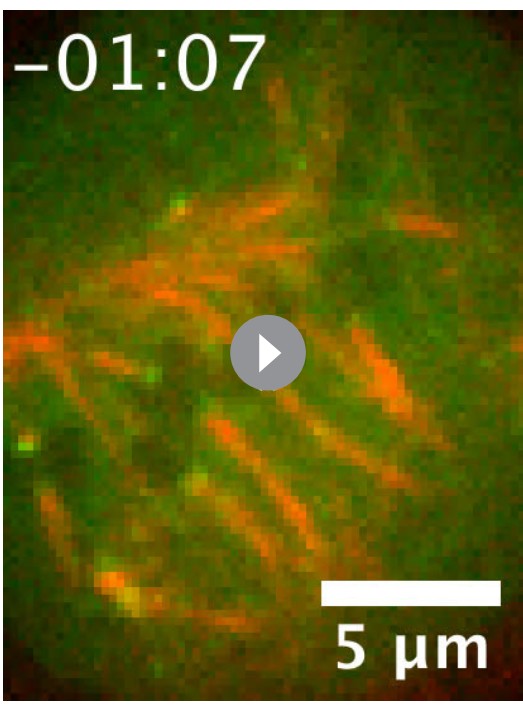

**Video 6.** Dynactin is not recruited to minus-ends in the absence of NuMA. See also *Figure 4A*. Live confocal imaging of a RPE1 cell in which NuMA has been knocked out using an inducible Cas9 system. In the absence of NuMA, 2xGFP-Arp1A does not localize to k-fiber minus-ends, and its recruitment is not detectable at new minus-ends (arrowheads) created by ablation ('X'). Note that mislocalized 2xGFP-Arp1A puncta diffuse randomly within the spindle, including within microns of the new minus-end, but do not remain there. Time is in min:s, with frame captured immediately following ablation set to 00:00. Plus-end of ablated k-fiber is marked by '*'. Scale bar, 5 µm.
DOI: https://doi.org/10.7554/eLife.29328.015

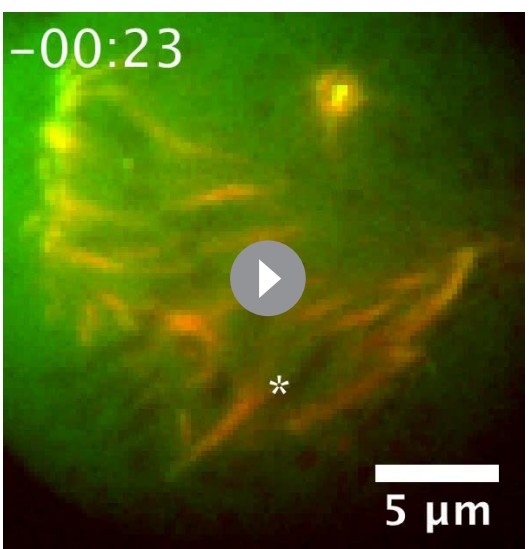

**Video 7.** The 'tail1+2A' region of NuMA's C-terminus is sufficient for minus-end localization. See also *Figure 6D*. Live confocal imaging of a RPE1 cell expressing GFP-C-tail1+2A in a NuMA knockout background. After ablation ('X'), GFP-C-tail1+2A is recruited to k-fiber minus-ends (arrowhead). The k-fiber stub slowly polymerizes, but its minus-end is not transported by dynein and remains detached from the spindle. Note that C-tail1+2A does not rescue spindle architecture; video shows one confocal slice of a three-dimensionally disorganized spindle. Also note that bright GFP signal from a neighboring interphase cell is noticeable on the left side of the video. Time is in min:s, with the frame captured immediately following ablation set to 00:00 s. Plus-end of ablated k-fiber is marked by '*'. Scale bar, 5 µm.
DOI: https://doi.org/10.7554/eLife.29328.020

Videos show a single spinning disk confocal Z-slice imaged over time and were formatted for publication using ImageJ and set to play at 35x relative to real time. Videos were corrected to play at a constant frame rate, even when the acquisition rate was not constant.

## Acknowledgements

We are grateful to Kara McKinley and Iain Cheeseman for inducible Cas9/CRISPR RPE1 cells, inducible DHC knockout HeLa cells, and helpful advice, and to Anna Akhmanova for GFP-CAMSAP1 and sharing unpublished CAMSAP1 data. We thank Takeo Usui and Ichiro Hayakawa for gatastatin, Alexey Khodjakov for PtK2 GFP-α-tubulin cells, and Duane Compton, Iain Cheeseman, Michael Davidson, and Trina Schroer for constructs (GFP-NuMA, GFP-Arp1A, mEmerald-MAPTau, and DsRed-CC1 + mCherry-p50, respectively). Many thanks to Richard McKenney, Ruensern Tan, Tim Mitchison, and the Dumont Lab for discussions and critical reading of the manuscript, to Meelad Amouzgar and Rocio Gomez for technical assistance, and to Jonathan Kuhn and Mary Elting for help with image analysis code.

## Additional information

### Funding

| Funder | Grant reference number | Author |
|---|---|---|
| National Institute of General Medical Sciences | DP2GM119177 | Sophie Dumont |
| National Cancer Institute | NRSA F31 | Christina L Hueschen |
| Chicago Community Trust | Searle Scholars Program | Sophie Dumont |
| Rita Allen Foundation | Rita Allen Scholars | Sophie Dumont |
| National Science Foundation | Graduate Research Fellowship | Christina L Hueschen |
| University of California, San Francisco | Moritz Heyman Discovery Fellowship | Christina L Hueschen |
| Pew Charitable Trusts | Pew Biomedical Scholars | Ke Xu |
| National Science Foundation | CHE-1554717 | Ke Xu |
| National Institute of General Medical Sciences | R00GM09433 | Sophie Dumont |
| National Science Foundation | Center for Cellular Construction, NSF 1548297 | Sophie Dumont |

The funders had no role in study design, data collection and interpretation, or the decision to submit the work for publication.

### Author contributions

Christina L Hueschen, Conceptualization, Data curation, Formal analysis, Funding acquisition, Investigation, Visualization, Writing—original draft, Writing—review and editing, Performed all experiments except STORM imaging; Samuel J Kenny, Visualization, Methodology, Performed STORM imaging and analyzed STORM data; Ke Xu, Software, Supervision, Funding acquisition, Methodology, Provided guidance and funding for STORM experiments; Sophie Dumont, Conceptualization, Resources, Supervision, Funding acquisition, Methodology, Project administration, Writing—review and editing

### Author ORCIDs

Christina L Hueschen (iD) http://orcid.org/0000-0002-3437-2895
Ke Xu (iD) http://orcid.org/0000-0002-2788-194X
Sophie Dumont (iD) http://orcid.org/0000-0002-8283-1523

### Decision letter and Author response

Decision letter https://doi.org/10.7554/eLife.29328.027
Author response https://doi.org/10.7554/eLife.29328.028

## Additional files

**Supplementary files**

• Transparent reporting form

DOI: https://doi.org/10.7554/eLife.29328.023

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
