## [Decision Letter]

Thank you for submitting your article "NuMA Targets Dynein to Microtubule Minus-Ends at Mitosis" for consideration by *eLife*. Your article has been favorably evaluated by Anna Akhmanova (Senior Editor) and three reviewers, one of whom is a member of our Board of Reviewing Editors. The following individual involved in review of your submission has agreed to reveal their identity: Reto Gassmann (Reviewer #2).

The reviewers have discussed the reviews with one another and the Reviewing Editor has drafted this decision to help you prepare a revised submission.

Summary:

In this manuscript, Hueschen and co-workers use microscopy, laser ablation and molecular genetic techniques to investigate NuMA's role in regulating dynein and dynactin within the mitotic spindle. Their findings include the observation that NuMA localizes rapidly to newly created microtubule minus ends, and that this behaviour does not appear to depend on dynein, dynactin, or known minus-end binders γ-TuRC, CAMSAP1, and KANSL1/3. Conversely, the localization of dynactin (and, by inference, dynein) does depend on NuMA. The experiments suggest a specific region of NuMA autonomously recognizes minus ends, thereby recruiting dynein activity there. This work provides interesting evidence regarding the hierarchy of minus end association between NuMA, dynein, and dynactin, while raising a number of mechanistic puzzles and experimental queries.

This manuscript exhibits rigor in quantifying the localization kinetics of microtubule minus-end-binding proteins and in using knockout cell lines to perform clean molecular replacement experiments. The results are well presented with sound statistical analysis and fully support the authors' conclusions. The central idea of the study – NuMA targets dynein motor activity specifically to microtubule minus ends, and not vice versa as previously thought – is an important novel insight that will be of great interest to motor biochemists, developmental biologists and cell biologists.

Essential revisions:

1) Minus-end recognition

In several places in the manuscript, it is stated that NuMA "recognizes" minus ends. To many, this will imply a direct interaction, which is not explicitly tested in this work. The authors state that previous in vitro experiments with related NuMA fragments have not shown specific minus-end localization. An in vitro microtubule-binding experiment with the exact 'C-tail1+2A' construct used in this work would greatly strengthen the authors conclusions. Alternatively, instances of "recognizes" should be softened to "localizes to" or another expression that does not imply direct binding.

Further, discussion of the Sedlin et al. (2016) study could be expanded on. For example, if the 1811-1985 region of NuMA localizes to plus and minus ends as that paper suggests, is it conceivable that cellular factors are required to prevent plus-end binding? The notion of microtubule curvature recognition by NuMA is also raised in that paper. The authors should comment on these points.

2) NuMA localization under conditions of dynein inhibition or knockout

To investigate whether NuMA is localized to minus ends by direct recruitment or dynein transport, dynein activity is perturbed by p50 overexpression, p150 overexpression, and knockout of the dynein heavy chain. However, the kinetics of NuMA recruitment are only probed in detail in the p50 overexpression case. This provides the interesting observation that NuMA can still localize to minus ends when dynein activity is perturbed, but apparently with slower kinetics, suggesting that both direct recruitment and transport may contribute to NuMA localization. However, as the kinetic traces are noisy, it would be important to show the same trend in at least one of the other dynein inhibition conditions (i.e. perform laser ablation experiments, ideally in the dynein heavy chain knockout cell line).

The key point here is to strengthen the conclusion that dynein transport alone does not target NuMA to minus ends.

3) Figure 1: NuMA is shown to reach the spindle poles before dynactin, suggesting it recruits dynein/dynactin, not the other way round. There is no measurement of dynein. The authors should explain why dynein was not looked at.

4) Figure 2 reports that a DHC KO still leads to Numa and dynactin localization. This suggests that dynactin can be localized by NuMA independently of dynein. An important control is to show that the dynein level is completely knocked down by Western blot.

5) Figure 4 shows how NuMA targets dynactin. It would be helpful to the reader to include the dynein localization (currently in Figure 4—figure supplement 1) in this main figure. It explains why the authors focus on dynactin localization and do not directly address dynein localization.

6) Gatasin treatment

Do the authors have data to establish how effectively γ-TuRC was removed from minus ends by this treatment? Spindle length measurements do not probe this directly.

7) Super-resolution microscopy

A control with another minus-end binder (e.g. CAMSAP) needs to be included to provide convincing evidence for "organized clusters of NuMA puncta", rather than a lawn of molecules along the lattice near the minus-end.

8) Title: NuMA Targets Dynein to Microtubule Minus-Ends at Mitosis

There is a disconnect between title and the experiments in the study, as the localization data pertain mainly to dynactin rather than dynein. "Dynein" should be replaced with "Dynein and Dynactin" or "dynein activity".

There may also be some confusion with the use of "Targets" as some will feel dynein naturally targets minus end via its motor activity. Replacing with "Recruits" or "Localizes" may avoid this confusion.

9) The Materials and methods suggest that T-tests were used for statistical comparisons. However, because multiple comparisons are made, ANOVA and a suitable post-hoc test seems more appropriate and avoids the risk of Type I errors with multiple T-tests.

10) A 2xGFP-Arp1A construct is generated. Please state how the functionality of this construct in the dynactin complex was verified.

11) Figure 1. As no EB1 signal is shown, the microtubule polarity is not clear in the example shown.

Also, please comment on the fidelity of EB1 labelling plus ends of acentrosomal microtubules i.e. do these microtubules ever have EB1 signal at both ends?

12) N-Tau experiments

The conclusion that dynein needs to be localized at the minus end, rather along the length of the microtubule, is based on use of an N-Tau construct. Please comment on caveats associated with this construct. For example, is the affinity between Tau and the microtubule load bearing? What is the expression level of the N-Tau construct compared to endogenous NuMA and other constructs?"

---

## [Author Response]

Essential revisions:1) Minus-end recognitionIn several places in the manuscript, it is stated that NuMA "recognizes" minus ends. To many, this will imply a direct interaction, which is not explicitly tested in this work. The authors state that previous in vitro experiments with related NuMA fragments have not shown specific minus-end localization. An in vitro microtubule-binding experiment with the exact 'C-tail1+2A' construct used in this work would greatly strengthen the authors conclusions. Alternatively, instances of "recognizes" should be softened to "localizes to" or another expression that does not imply direct binding.Further, discussion of the Sedlin et al. (2016) study could be expanded on. For example, if the 1811-1985 region of NuMA localizes to plus and minus ends as that paper suggests, is it conceivable that cellular factors are required to prevent plus-end binding? The notion of microtubule curvature recognition by NuMA is also raised in that paper. The authors should comment on these points.

We appreciate the feedback that our language in certain places implied knowledge of a direct interaction between NuMA and minus-ends. We have carefully reviewed the text with this concern in mind and have softened our language to better convey our intended message: that our data do not distinguish between direct or indirect recruitment of NuMA to minus-ends (subsection “NuMA localizes to minus-ends independently of dynein”, last paragraph; subsection “NuMA is required for dynein activity and dynactin localization at minus-ends”, last paragraph; subsection “NuMA recognizes localizes to minus-ends independently of known minus-end binders”; subsection “NuMA function requires both minus-end-recognition and dynactin-recruitment modules”, second paragraph; subsection “NuMA targets dynactin to minus-ends, spatially regulating dynein activity at mitosis”, last paragraph; Figure 6 legend). We also thank the reviewers for the suggestion to expand our discussion of the Seldin et al. (2016) study. We have done so in the last paragraph of the subsection “NuMA targets dynactin to minus-ends, spatially regulating dynein activity at mitosis”.

2) NuMA localization under conditions of dynein inhibition or knockoutTo investigate whether NuMA is localized to minus ends by direct recruitment or dynein transport, dynein activity is perturbed by p50 overexpression, p150 overexpression, and knockout of the dynein heavy chain. However, the kinetics of NuMA recruitment are only probed in detail in the p50 overexpression case. This provides the interesting observation that NuMA can still localize to minus ends when dynein activity is perturbed, but apparently with slower kinetics, suggesting that both direct recruitment and transport may contribute to NuMA localization. However, as the kinetic traces are noisy, it would be important to show the same trend in at least one of the other dynein inhibition conditions (i.e. perform laser ablation experiments, ideally in the dynein heavy chain knockout cell line).The key point here is to strengthen the conclusion that dynein transport alone does not target NuMA to minus ends.

As suggested, we have performed laser ablation experiments in the inducible dynein heavy chain (DHC) knockout HeLa cell line. We measured GFP-NuMA recruitment to new minus-ends with and without deletion of DHC, and the data show the same trends as with and without p50 overexpression in PtK2 cells (Figure 2). NuMA is robustly recruited to HeLa k-fiber minus-ends even after DHC deletion, but recruitment is slightly delayed. We have added these data to Figure 2—figure supplement 1. We thank the reviewers for strengthening the manuscript with this suggestion.

3) Figure 1: NuMA is shown to reach the spindle poles before dynactin, suggesting it recruits dynein/dynactin, not the other way round. There is no measurement of dynein. The authors should explain why dynein was not looked at.

We have added an explanation as suggested (subsection “Dynactin and NuMA display mitosis-specific, steady-state binding at microtubule minus-ends”, second paragraph). Thank you for this feedback. In brief, high cytoplasmic levels of dynein result in variable and low signal-to-background imaging of the motor on microtubules. This precluded accurate measurements of dynein’s dynamics.

4) Figure 2 reports that a DHC KO still leads to Numa and dynactin localization. This suggests that dynactin can be localized by NuMA independently of dynein. An important control is to show that the dynein level is completely knocked down by Western blot.

We agree with the reviewers and have added a western blot demonstrating dynein depletion after inducible DHC KO to Figure 2—figure supplement 1. Furthermore, dynein depletion was always verified in individual cells analyzed – by spindle phenotype or, whenever possible, by immunofluorescence labeling of dynein.

5) Figure 4 shows how NuMA targets dynactin. It would be helpful to the reader to include the dynein localization (currently in Figure 4—figure supplement 1) in this main figure. It explains why the authors focus on dynactin localization and do not directly address dynein localization.

We thank the reviewers for this feedback and have moved the dynein localization into the main Figure 4, as suggested.

6) Gatasin treatmentDo the authors have data to establish how effectively γ-TuRC was removed from minus ends by this treatment? Spindle length measurements do not probe this directly.

To more directly test how effectively gatastatin blocks γ-TuRC’s interaction with minus-ends, we performed a microtubule regrowth assay in the presence of 30 μM gatastatin or equivalent DMSO (as in Chinen et al., Nat Comm 2016). After cold treatment, 30 μM gatastatin dramatically blocked γ-TuRC’s ability to bind α-tubulin to nucleate new microtubules (bottom panels, below). Gatastatin did not depolymerize existing microtubule networks (top panels), demonstrating its specificity in binding γ-tubulin, not α- or β-tubulin. We thank the reviewers for suggesting this important control and have included these data in Figure 5—figure supplement 1. Details of the re-growth assay have been added to the manuscript’s Materials and methods.

7) Super-resolution microscopyA control with another minus-end binder (e.g. CAMSAP) needs to be included to provide convincing evidence for "organized clusters of NuMA puncta", rather than a lawn of molecules along the lattice near the minus-end.

We thank the reviewers for this suggestion. We have performed 3D STORM of CAMSAP at PtK2 k-fiber minus-ends created by ablation, as done for NuMA in Figure 5. In Author response image 1, we show examples of k-fiber minus-ends (white boxes) created by ablation and imaged by 3D STORM after immunofluorescence labeling of NuMA (top) or CAMSAP (bottom). In middle panels, structures are colored according to position in the Z-axis. Cartoons depict orientation of k-fiber and minus-end. Like NuMA, CAMSAP forms organized clusters of puncta, and distances between neighboring ‘nodes’ of both CAMSAP and NuMA (e.g., ‘*’) are ~50-150 nm (*n* = 32 nodes, 4 ablations for NuMA; *n* = 28 nodes, 3 ablations for CAMSAP). This spacing is consistent with measured spacing between individual microtubules within PtK2 k-fibers (McDonald et al., J. Cell Biol. 1992). These data support the idea of “organized clusters of NuMA puncta, rather than a lawn of molecules along the lattice near the minus-end”.

Unfortunately, limited signal from these minus-ends structures meant that we could only detect NuMA or CAMSAP clearly when labeled with the high-photon-yield and low-duty-cycle dye Alexa 647; this precluded two-color imaging of NuMA and CAMSAP co-localization in the same cell. In addition, while the large size and flat shape of mitotic PtK2 (rat kangaroo) cells made STORM experiments possible, commercially available CAMSAP1 antibodies raised against human antigen did not label any PtK2 proteins. The images in Author response image 1 show immunofluorescence staining with a CAMSAP2 antibody (17880-1-AP; Proteintech). In human cells (e.g., RPE1), this CAMSAP2 antibody does not label spindles, as human CAMSAP2 is phosphorylated at mitosis and does not interact with microtubules (Jiang et al., Dev Cell2014). Although these data support our claims, we do not include them in the manuscript due to the above limitations – and to avoid potential confusion looking forward.

8) Title: NuMA Targets Dynein to Microtubule Minus-Ends at MitosisThere is a disconnect between title and the experiments in the study, as the localization data pertain mainly to dynactin rather than dynein. "Dynein" should be replaced with "Dynein and Dynactin" or "dynein activity".There may also be some confusion with the use of "Targets" as some will feel dynein naturally targets minus end via its motor activity. Replacing with "Recruits" or "Localizes" may avoid this confusion.

We appreciate this feedback and have changed the title accordingly: NuMA Recruits Dynein Activity to Minus-Ends at Mitosis.

9) The Materials and methods suggest that T-tests were used for statistical comparisons. However, because multiple comparisons are made, ANOVA and a suitable post-hoc test seems more appropriate and avoids the risk of Type I errors with multiple T-tests.

We thank the reviewer for this suggestion. We conducted ANOVA analyses on all data in which multiple comparisons are made: Figure 1 (Arp1A, NuMA, and CAMSAP1 recruitment timescales to minus-ends after ablation); Figure 5 (NuMA recruitment timescales after inhibition or deletion of minus-end binding proteins); and, Figure 6 (spindle organization after rescue with NuMA truncations). Results are reported in Figure panels 1G, 5C, 6C and their corresponding figure legends, as well as below.

Figure 1: To compare the timescales of Arp1A, NuMA, and CAMSAP1 recruitment to new minus-ends, we used a one-way ANOVA, which revealed a statistically significant difference between groups (*F*(2,38) = 9.26, *p* = 0.0005). A Tukey post hoc test showed a significant difference between recruitment timescales for Arp1A vs. NuMA (*p* = 0.03) and Arp1A vs. CAMSAP1 (*p* = 0.001). NuMA vs. CAMSAP1 differed at *p* = 0.10.

Figure 5 one-way ANOVA showed no statistically significant difference between NuMA recruitment timescales after gatastatin treatment, CAMSAP1 knockout, or KANSL1 knockout: *F*(3,42) = 0.45, *p* = 0.72.

Figure 6 one-way ANOVA revealed statistically significant differences in the percentage of cells showing focused, bipolar spindle architecture after rescue experiments with different NuMA truncations (*F*(3,10) = 267, *p* <0.00001). A Tukey post hoc test showed a significant difference between recruitment timescales for FL or N-C vs. C (*p* = 0.001) and FL or N-C vs. N-Tau (*p* = 0.001).

10) A 2xGFP-Arp1A construct is generated. Please state how the functionality of this construct in the dynactin complex was verified.

The 2xGFP-Arp1A construct localized correctly within the spindle (e.g., at poles and brightly at kinetochores without complete end-on attachments), just like 1xGFP-Arp1A, and it did not perturb spindle organization. We have added a sentence stating this in the Materials and methods (subsection “Plasmids”).

11) Figure 1. As no EB1 signal is shown, the microtubule polarity is not clear in the example shown.Also, please comment on the fidelity of EB1 labelling plus ends of acentrosomal microtubules i.e. do these microtubules ever have EB1 signal at both ends?

We thank the reviewers for this feedback. We have changed the corresponding text to make it clear that Figure 1, not 1A, provides the information about microtubule polarity (subsection “Dynactin and NuMA display mitosis-specific, steady-state binding at microtubule minus-ends”, first paragraph). Interestingly, we see EB1 label plus-ends, not minus-ends, of single acentrosomal microtubules with high fidelity at mitosis. (Small bundles of microtubules, on the other hand, sometimes show EB1 labeling at both ends, presumably because they contain microtubules of mixed polarity.) In agreement with this specificity, we have never been able to detect evidence of microtubule polymerization at minus-ends after ablation in mitosis, at least in PtK2 cells. We have added a clarifying note on this topic to the Materials and methods (subsection “Data Analysis”, first paragraph).

12) N-Tau experimentsThe conclusion that dynein needs to be localized at the minus end, rather along the length of the microtubule, is based on use of an N-Tau construct. Please comment on caveats associated with this construct. For example, is the affinity between Tau and the microtubule load bearing? What is the expression level of the N-Tau construct compared to endogenous NuMA and other constructs?"

We have added a discussion of this caveat: “we cannot formally exclude […] that while N-Tau recruits dynactin (Figure 6), it cannot effectively couple to dynein.” In addition, other experimental and computational studies (e.g., Elie et al., Sci. Rep. 2015; Ahmadzadeh et al., Biophys. J.2015) suggest that Tau’s interaction with the microtubule can bear load.

To quantify the expression level of the GFP-N-Tau construct compared to full-length (FL) GFP-NuMA, we measured the average GFP intensity within the spindle region (normalized to cytoplasmic background intensity) for a confocal slice through the spindle center, for all cells used in our analysis of spindle organization (Figure 6). Unfortunately, more ideal comparisons of integrated intensity within the whole spindle volume were not possible, given how these data were acquired. However, these average intensity measurements (Author response image 2) suggest that N-Tau expression was comparable or higher than full-length NuMA. Each data point represents an individual cell; *n* = 19 for FL NuMA and *n* = 16 for N-Tau. Thus, N-Tau’s inability to rescue spindle organization is not due to low protein expression. Comparison to endogenous levels was not possible, as the polyclonal NuMA antibody binds with unknown stoichiometry to N-Tau vs. NuMA. Due to these limitations, we do not report this data in the manuscript.

**Author response image 2. respfig2:**